# Developing a crop- wild-reservoir pathogen system to understand pathogen evolution and emergence

Mark McMullan[1]*, Lawrence Percival-Alwyn[2], Gemy G Kaithakottil[1], Laura-Jayne Gardiner[3], Rowena Hill[1], Hélène Yvanne[1], Michelle Grey[1], Kevin Sawford[4], Sabrina Jaye Ward[1], Ross Low[1], Sally D Warring[1], Darren Heavens[1], Ned Peel[1], Jakob Kroboth[1], Mark Stevens[4], David Swarbreck[1], Matt D Clark[5], Neil Hall[1]

[1]The Earlham Institute Norwich Research Park, Norwich, United Kingdom; [2]National Institute of Agricultural Botany, Cambridge, United Kingdom; [3]IBM Research Europe, Warrington, United Kingdom; [4]British Beet Research Organisation, Norwich, United Kingdom; [5]Department of Life Science, The Natural History Museum, London, United Kingdom

*For correspondence:
mark.mcmullan@earlham.ac.uk

Competing interest: The authors declare that no competing interests exist.

**Abstract** Crop pathogens reduce yield and contribute to global malnourishment. Surveillance not only detects presence/absence but also reveals genetic diversity, which can inform our understanding of rapid adaptation and control measures. An often neglected aspect is that pathogens may also use crop wild relatives as alternative hosts. This study develops the beet (*Beta vulgaris*) rust (*Uromyces beticola*) system to explore how crop pathogens evolve to evade resistance using a wild reservoir. We test predictions that crop selection will drive virulence gene differentiation and affect rates of sex between crop- and wild-host rust populations. We sequenced, assembled, and annotated the 588 Mb beet rust genome, developed a novel leaf peel pathogen DNA extraction protocol, and analysed genetic diversity in 42 wild and crop isolates. We found evidence for two populations: one containing exclusively wild-host isolates; the other containing all crop-host isolates, plus five wild isolates. Effectors showed greater diversity in the exclusively wild population and greater differentiation between populations. Preliminary evidence suggests the rates of sexual reproduction may differ between populations. This study highlights how differences in pathogen populations might be used to identify genes important for survival on crops and how reproduction might impact adaptation. These findings are relevant to all crop-reservoir systems and will remain unnoticed without comparison to wild reservoirs.

## Editor's evaluation

This important study addresses the question of how pathogen reservoirs can play a role in the emergence of new virulences, using beets and Uromyces beticola as a pathosystem. The authors find convincing evidence for higher signals of diversity and differentiation in effector genes and significant differences in reproductive rates between two pathogen populations that occur either on wild or on wild and domesticated beets. These findings have the potential for developing approaches that reduce the evolution of pathogen resistance in crops.

## Introduction

The world's human population is projected to increase to 9.9 billion by 2050, yet two billion people are currently malnourished. With over 20% of crops being lost to disease annually, there is an opportunity to increase food production by improving crop disease tolerance/resistance (*Bruinsma and Bruinsma, 2012*; *Savary et al., 2012*). Crop protection by genetic resistance is characterised by short disease-free periods, or a lag while pathogens adapt (*Brown, 2015*; *Karasov et al., 2020*). Once resistance is broken, emergent diseases spread quickly, often clonally, through the host population. New sources of resistance take several years to develop and wild crop relative species are being explored as potential reservoirs of resistance gene diversity (*Arora et al., 2019*; *Karasov et al., 2020*). These wild crop relatives are also reservoirs for pathogens, and once we accept that crop pathogens may be exchanging genetic variation with reservoirs occupying different environments, we introduce the importance of understanding these as sources of novel pathogen genetic variation. Perhaps more importantly, crop-wild pathogen populations present an opportunity to identify genetic variation specifically adapted to crops (*McCann, 2020*). Yet much of the work on crop pathogen emergence has not accounted for this wild reservoir of genetic diversity, and those that have consider one-off invasion scenarios, as opposed to understanding ongoing evolution through adaptation and gene flow (e.g. *Grünwald et al., 2016*; *Martin et al., 2016*; *Martin et al., 2014*, but see, *McMullan et al., 2018*).

Gene flow in its broadest sense occurs at different rates. At one end of the spectrum, we have single hybridisation/introgression events and, at the other, contemporary gene flow. Hybridisation events have been implicated in the generation and success of plant pathogens such as Dutch Elm Disease (*Brasier and Kirk, 2010*), wheat blast (*Rahnama et al., 2023*) and *Zymoseptoria pseudotritici* (*Stukenbrock et al., 2012*), as well as human pathogens such as malaria, *Plasmodium falciparum* (*Galaway et al., 2019*), and sleeping sickness, *Trypanosoma brucei* (*Goodhead et al., 2013*). An example of an intermediate level of genetic exchange is seen in the brassica white rust pathogen (*Albugo candida*) that can infect over 200 plant species and in which host specialised lineages diverge but rare introgression events distribute diversity between lineages (*Jouet et al., 2018*; *McMullan et al., 2015*). Recombination within a diverse genetic reservoir has also been implicated in the generation of the rice blast pathogen, a devastating crop pathogen of global significance (*Latorre et al., 2020*). However, while these rarer processes are important to understand the generation of novel pathogens (*McCann, 2020*; *Stukenbrock, 2016*), the ongoing adaptation of crop pathogens to novel resistances may be better understood by using population genetics to consider their access to reservoirs of genetic diversity (*Gladieux et al., 2015a*). Developments in genomic surveillance technologies such as Air-seq, may reduce the resource required to survey plant pathogens (*Giolai et al., 2024*; *Peers et al., 2024*; *Weisberg et al., 2021*). Advances to our understanding of pathogen adaptation and emergence will benefit from replicated invasion experiments, and crop landscapes provide that replication. Crop domestication has bottlenecked plant genetic diversity in order to homogenise traits (*Kumar et al., 2021*). Infection of genetically depauperate hosts will favour specific pathogen gene diversity and selection at replicated crop environments can facilitate this understanding.

Rust fungi make up some of the most economically devastating crop pathogens, infecting crops of global significance such as wheat (*Schwessinger et al., 2020*), soybean (*Loehrer et al., 2014*), and coffee (*Porto et al., 2019*). They are also ideal systems for exploring host adaptation and the crop-reservoir pathogen populations because they are known for their complex life cycles, which include changes in ploidy, partitioned sexual, clonal, overwintering, and dispersal stages (*Kristoffersen et al., 2018*), but primarily because of their obligate association with the living host (*Figueroa et al., 2020*). In an agricultural context, the main infection phase is clonal, where urediniospores are found erupting from pustules on the surface of leaves (*Figure 1A*). Each spore contains two haploid nuclei (dikaryotic). This clonal phase can continue indefinitely on susceptible hosts but can also alternate with a seasonal sexual stage, often on a wild host species (*Figueroa et al., 2020*). While there are exceptions, the general principle is that mutation increases heterozygous polymorphism between dikaryotic content (as well as within the population) in the clonal phase, and recombination shuffles this variation in the sexual phase (*Figueroa et al., 2020*; *Schwessinger et al., 2020*). The rare sexual reproduction on a wild host may be a key determinant in bouts of adaptation. In most cases, the sexual phase of the life cycle occurs on a different host plant, termed heteroecism (as opposed to autoecism). In the case of wheat stem rust, this alternate host is barberry, and the presence of this alternate host has been associated with increased rust virulence in Europe and the US for hundreds of years (*Barnes et al., 2020*).

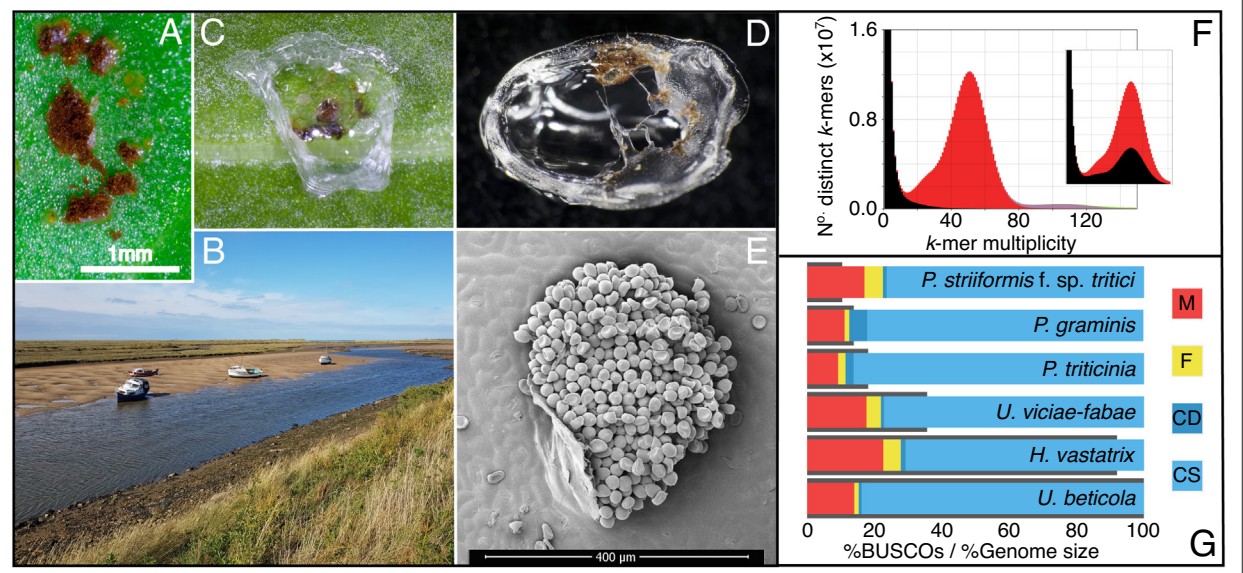

**Figure 1.** Zooming in on beet rust, pustules from wild and crop hosts are peeled and sequenced. (**A**) *U. beticola* pustules on sugar beet. (**B**) Sea beets in their natural setting along an estuary, located bottom right and middle of the bank. Leaves are clipped and brought back into the lab. (**C & D**) Pustules are covered with 5 µl peel solution which dries and is peeled off for DNA extraction, library preparation, and sequencing. (**E**) An electron micrograph image of a rust pustule, isolate urediniospores are visible. (**F**) The rust genome *k*-mer spectra is a histogram of the number of *k*-mers (in the reads) found at a given multiplicity (or depth) at *k*-mers in the assembly. *K*-mers present in the reads but absent in the assembly are plotted in black, present once in the assembly in red, then purple for twice. The inset plot shows the *k*-mer distribution where all contigs without a blast hit are removed. The black peak within the main distribution indicates that real *U. beticola* content has been removed and should not be filtered from the assembly. (**G**) BUSCO completeness scores for our *U. beticola* (588.0 Mbp) genome in comparison to *Hemileia vastatrix* (541.2 Mbp), *Uromyces viciae-fabae* (209.5 Mbp), *P. triticinia* (106.6 Mbp), *Puccinia graminis* (81.5 Mbp) and *P. striiformis* f. sp. *Tritici* (61.4 Mbp). Colours indicate Missing, Fragmented, Complete and Duplicated, and Complete and Single copy BUSCOs. Genomes are in size order, represented as grey bars outlining BUSCO scores. The *U. beticola* genome, while large, has comparatively low levels of missing, fragmented, and duplicated BUSCO content.

The online version of this article includes the following figure supplement(s) for figure 1:

**Figure supplement 1.** Combined interspersed repeat content relative to genome size.

Alternate hosts act as centres of rust genetic diversity and recombination, and these are important forces for adaptation which operate outside of crop-infecting populations. However, these processes are difficult to study, either because the wild hosts are peppered throughout a global range and difficult to locate, and/or they are in regions that are difficult to access. For instance, in wheat yellow (stripe) rust, the centre of genetic diversity and recombination are in Nepal, Pakistan, and China, and yet their largely clonally sustained populations expand to every continent except Antarctica (*Ali et al., 2014*). The present study is an attempt to resolve this obstacle by developing a system, the crop-wild beet rust system – sugar beet (*B. vulgaris* subsp. *vulgaris*), sea beet (*Beta vulgaris* subsp. *maritima*), and beet rust (*Uromyces beticola*) – which is ideally suited to study crop-wild pathogen evolution. Sugar beet is one of the most recently domesticated crop species (circa 200 y, *Dohm et al., 2014*), which is grown for sugar production and is being developed as a biofuel throughout Europe and the USA (*Panella, 2010*). Domesticated in Europe, it is often grown alongside coastally distributed sea beet, which remains a source of genetic variation for sugar beet breeding (*Cauwer et al., 2012*; *Leys et al., 2014*; *Sandell et al., 2022*; *Figure 1B*). *Kristoffersen et al., 2018* present the beet rust disease cycle, showing both sugar and sea beet are impacted by the autoecious rust pathogen, *U. beticola*. Being an obligate biotroph *U. beticola* must survive on wild hosts between sugar beet crops cycles. Despite living on a single host species, the beet rust lifecycle contain urediospore (n+n) and teliospore (2 n; and subsequent basidiospore (n), pycniospore (n) and aeciospores (n+n)) stages (*Kristoffersen et al., 2018*). It is the asexual urediospores that can cyclically reinfect, and sexual basidiospores that are important in the consideration of the impact of a partitioned lifecycle for invasion and evolution. Although relatively little is known about resistance mechanisms in beet species, one resistance locus, the Rz2 locus responsible for rhizomania resistance, has been well described (*Rodríguez et al.,*

*2019*). Microsatellite analyses of wild sea beet populations demonstrate high levels of allelic diversity and heterozygosity, associated with outbreeding, which is corroborated at the genome scale (*Fénart et al., 2008*; *Ribeiro et al., 2016*; *Stevanato et al., 2001*; *Sandell et al., 2022*).

Here, we explore the utility of the beet-rust system as a model to understand pathogen adaptation to agricultural environments. We have produced the first annotated *U. beticola* genome assembly and characterised population genetic diversity of rust isolates from sugar and sea beets collected across the east of England. We developed a DNA peel extraction protocol to favourably extract pathogen DNA relative to host DNA and re-sequenced 42 rust isolates (24 from sugar beet hosts and 18 from sea beets; ~370 km of coastline). Our predictions were that (A) isolates would partition into populations separated by the wild and crop host divide (allowing for some cross-infection) (B) genes important for success in these environments would be more genetically differentiated between populations, and that (C) the rate of clonal to sexual reproduction would be elevated in the crop pathogen population. Genes important for adaptation to the agricultural environment likely have many functions. However, we reason that host genetic diversity is one of the largest discriminating factors, and therefore we focused on pathogen genetic diversity at putative host interaction genes (effectors). Effector candidates can be readily identified using their secretion signals in the gene annotation process. Most importantly, these genes are known to be at the centre of the virulence-resistance arms race, where the right combination of effectors will increase the virulence of an isolate, and yet host recognition of a single effector will prevent infection (*Stukenbrock and McDonald, 2009*). We find that (A) rust isolates from all crop hosts belong to a single genetic cluster, which also included five northern wild host isolates, indicating a potential invasion source. The remaining wild-host isolates fall in an exclusively wild clade. We find that (B) this wild reservoir clade has especially high diversity at effector genes, which are also more differentiated than non-effectors. We find that (C) the rate of clonal to sexual reproduction differs between populations, although this signal is in opposite directions based on genotypic and linkage-based analyses. Our results are by no means final, but they do bring us closer to understanding ongoing processes that generate novel virulent pathogen lineages. From a surveillance perspective, these methods have implications for pathogen invasion and emergence, and our ability to target treatment and reduce chemical inputs (*Richard et al., 2022*).

## Results

### *U. beticola* genome assembly

We assembled the *U. beticola* genome (588 Mbp) using short-read data into 19,690 contigs with a genome N50 of 74 Kbp, the largest contig being 554 Kbp (*Table 1*). Rust genomes are often problematic to assemble because of the dikaryotic (n+n) content of the uredospore life stage (*Aime et al., 2017*; *Kristoffersen et al., 2018*). *K*-mer sampling read content is a method of ascertaining whether the content in the sequencing reads is reflected in the assembly they produce (*Mapleson et al., 2016*). Divergent haplotypic content would be evident as a double peak in the *k*-mer distribution of the reads (e.g. see *Figure 1* in ref *Loehrer et al., 2014*). We did not observe this signal characteristic of divergent haplotypic (heterozygous) content and instead, our plots suggest our assembly to have relatively low heterozygosity, misassembles or frame-shifts (*Mapleson et al., 2016*; *Figure 1F*). The

**Table 1.** Summary assembly statistics for *Uromyces beticola* (EI v1.1) genome.

| Mean | 29,880.49 |
| --- | --- |
| Median | 11,829 |
| Min | 1000 |
| Max | 554,123 |
| N50[length] | 294,203,835 |
| N50[value] | 74,005 |
| L50 | 2309 |
| Total length | 588,346,853 |
| Total sequences | 19,690 |

spectra also shows that rare *k*-mers (suspected errors in the reads, in black) are not found in the main distribution of the assembly. The content present once (in red) centres around the sequencing depth (~50 x) with a slight heterozygous shoulder (~25 x). To assess whether contigs that did not retrieve a blast hit were assembly artifacts, we removed them and replotted the *k-mer* spectra. We found that this removed content from the main part of the distribution, suggesting that these contigs are genuine and so they were retained (*Figure 1F*).

To assess completeness of our *U. beticola* assembly and annotation, we used core gene presence in comparison to a range of other rust assemblies of smaller and equivalent size (*Cantu et al., 2011*; *Link et al., 2014*; *Porto et al., 2019*; *Figure 1G*). BUSCO (*Simão et al., 2015*) genome completeness places our *U. beticola* genome within the level of other rust assemblies (85.1% of the complete BUSCOs; 84.0% single copy). Moreover, the level of duplicated BUSCOs is low in comparison to other rust assemblies where divergent dikaryotic content may be more problematic (*Figure 1G*).

The *U. beticola* genome annotation identified 9148 protein-coding genes (17,591 transcripts) with a mean transcript length of 2058 bp (6.4 exons per gene) and a mean coding sequence (CDS) length of 1240.9 bp (spliced from transcript in the annotation; see also *Supplementary file 1*). A large proportion of genes were annotated as transposable elements (6464 TEs) within this repeat-rich genome, in which low complexity and interspersed repeats represent 89.96% of the genome (see *Supplementary file 1*). Our reassessment of published rust genomes shows that this level of repeat content is consistent with such a large rust genome (*Figure 1—figure supplement 1*). Signal peptide information was used to define the secretome and then 225 putative effectors were identified using EFFECTORP2.0 (*Sperschneider et al., 2018*).

## *U. beticola* is differentiated into two UK populations

Sugar beet cultivation in the UK is based around the four sugar processing plants in the east of England. We analysed genome-wide SNP diversity of *U. beticola* isolated from wild coastal sea beets and (agricultural) sugar beets from this region. We sequenced 46 isolates and, after quality control, 42 (mean depth = 18.9; SD = 5.5) were from either a wild sea beet (n=18) or a sugar beet crop (n=24; *Appendix 1—table 1*). Using 1.87 million SNPs, we assessed genetic diversity at three levels. First, we used principal component analysis (PCA) and then discriminant analysis of principle components (DAPC) to identify structure in the distribution of genetic variation among isolates. PCA was run on 307,041 linkage pruned SNPs and represented as a scatter and a neighbour joining (NJ) tree. Isolates differentiate into two clusters across the first principal component (PC1, *Figure 2A*) which separates a group containing only wild-infecting isolates from another group containing all crop-infecting isolates plus five northern wild isolates. A neighbour joining tree is not able to accurately depict relationships among recombining isolates. However, the NJ tree does reflect populations partitioned across PC1 in addition to different patterns of diversification in both clades, which is consistent with clonality and rapid spread in the agricultural clade (*Figure 2A and B*). The five wild isolates found in the agricultural clade are not found together within the same subclade. DAPC is a method that utilises data transformation by PCA as a prior step to discriminant analysis, is not biased by linked sites, and is able to partition in the presence of clonally related isolates. Using 1.87 million SNPs, all 41 PCs were retained to determine using the lowest Bayesian information criterion (BIC), that isolates are partitioned into two clusters (see *Figure 2C*, Appendix 2). Fifteen PCs were used in the DAPC analysis, and this confirms that clusters are entirely disentangled across a single discriminant function (*Figure 2C*), with individual membership of each cluster equal to one (*Figure 2D*).

Clustering analyses show that the exclusively wild population spans three coastal sites (~80 km) and yet isolates from crop beets within this region are found in the other population, which contains all crop isolates (see Orford in *Figure 3A*; *Supplementary file 2*). However, this agricultural clade, containing all 24 crop isolates, does include five isolates from the two northernmost wild sites (*Figure 3*, asterisks).

To assess the impact of population membership on the rate of sexual reproduction, we used genetic diversity among isolates at gene CDS regions in a neighbour-net network (which allows branch fusion; *Figure 3B*) in addition to admixture using sparse non-negative matrix factorization (sNMF) clustering (LEA; *Frichot and François, 2015*; *Figure 3C*; Appendix 2). The neighbour-net network again shows differentiation of an exclusively wild population of isolates from three sites in the southeast of England (*Figure 3A and B*; Appendix 2). It is noteworthy that isolates from the five northern wild beets are

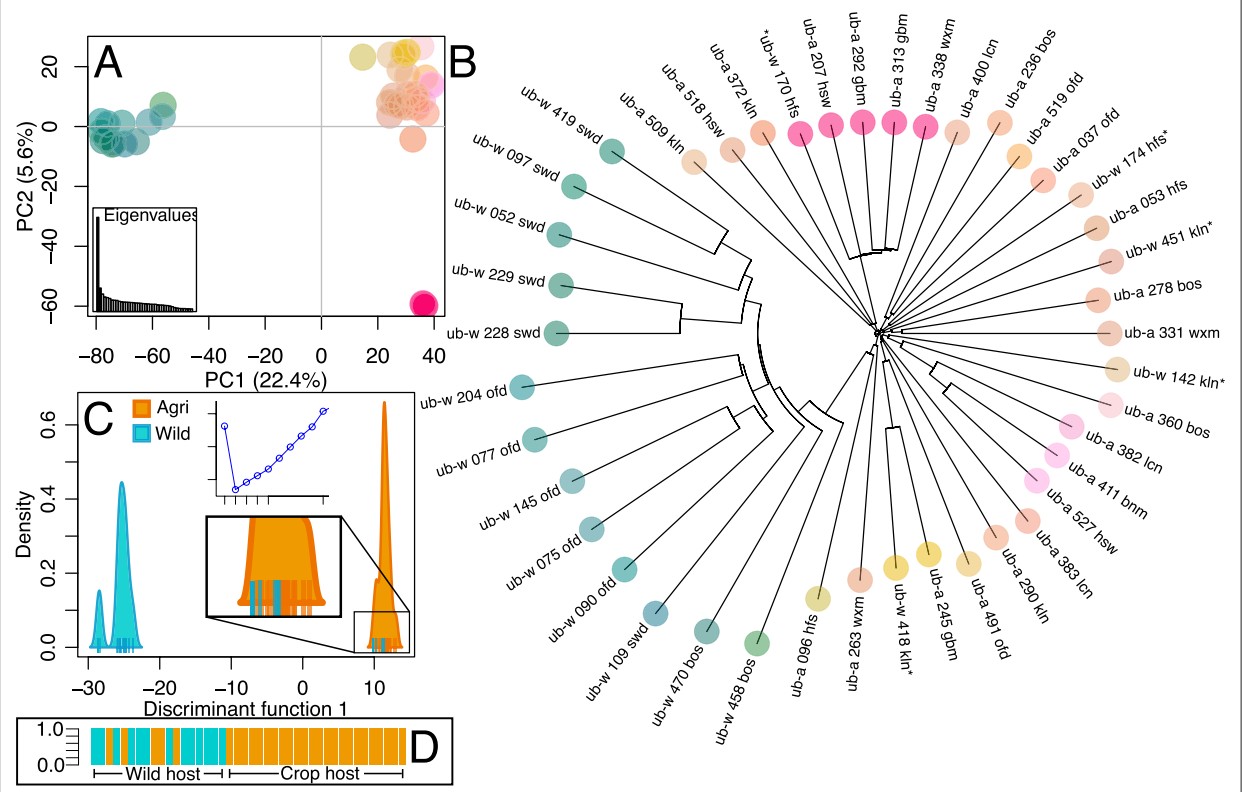

**Figure 2.** Population structure of wild and crop isolates. (**A**) Principal Component Analysis of 300 thousand linkage-pruned SNPs. Principle Component 1 accounts for 22.4% of variance and divides isolates into two groups, the first (green-blue) contains only wild-infecting isolates, from southern sites. The second group (orange-pink), contains all crop-infecting isolates plus five northern wild sampled isolates. Five isolates are apart from the crop-infecting group on PC2 (5.6%). Only one of which is a wild-infecting isolate (ub-a_207_hsw ub-a_292_gbm ub-a_313_gbm ub-a_338_wxm ub-w_170_hfs). Isolates are coloured according to their position on each axis. Eigenvalues are represented, inset bottom left. (**B**) Neighbour Joining tree (coloured according to PCA) reflects the relationship in the PCA and shows that northern wild isolates (asterisks) are not more closely related to each other than they are to other crop-infecting isolates. (**C**) Discriminant analysis of principal components (DAPC) of 1.87 million SNPs. Bayesian information criterion (BIC) shows that ($k$=2) two clusters best describe the data (inset top: x=No. clusters, y=BIC) and so isolates are plotted over a single discriminant function (DF1). Isolate ticks are coloured by host sampling group (wild = cyan, crop = orange) and distributions are coloured by predominant membership to indicate a population containing only wild isolates and a population containing all agricultural isolates along with 5 northern wild isolates. Groups are coherent between PCA and DAPC analyses as well as being entirely disentangled on the DAPC plot. (**D**) The probability of the strain belonging to each group is equal to 1, ordered by host type and coloured by population (agri = orange, wild = cyan).

again distributed throughout the clade of crop isolates, as opposed to clustering together. The fact that isolates from each site are distributed across the crop clade, suggests high levels of dispersal. With respect to wild and crop pathogen populations, dispersal may be bidirectional but is ultimately from a presumed wild source, as crop beets are not in the ground year-round but are required by this obligate biotroph. The exclusively wild isolates are clearly separated from the crop isolates, and reticulated patterns in the network suggest an impact of recombination which is found to a greater extent within the exclusively wild clade of isolates.

Admixture analysis is used here to assess whether the five northernmost isolates, that were sampled on a wild host but found within the crop-infecting lineage, carried any signal of hybridisation. We used an admixture methodology that does not assume Hardy–Weinberg equilibrium (*Frichot et al., 2014*), which is appropriate for species that can clonally reproduce. The methodology estimates population structure and admixture using sparse nonnegative matrix factorization (sNMF within LEA Frichot and François, 2015). We did not find a signal of admixture in our five northern wild-infecting isolates to distinguish them from the other crop-infecting isolates (*Figure 3C*).

We went on to explore the impact of increasing $k$ values, specifically on the assignment and admixture of the five northern wild sampled individuals. They do not consistently partition with each other exclusively from the agricultural population (Appendix 2). Machine learning methods were also not

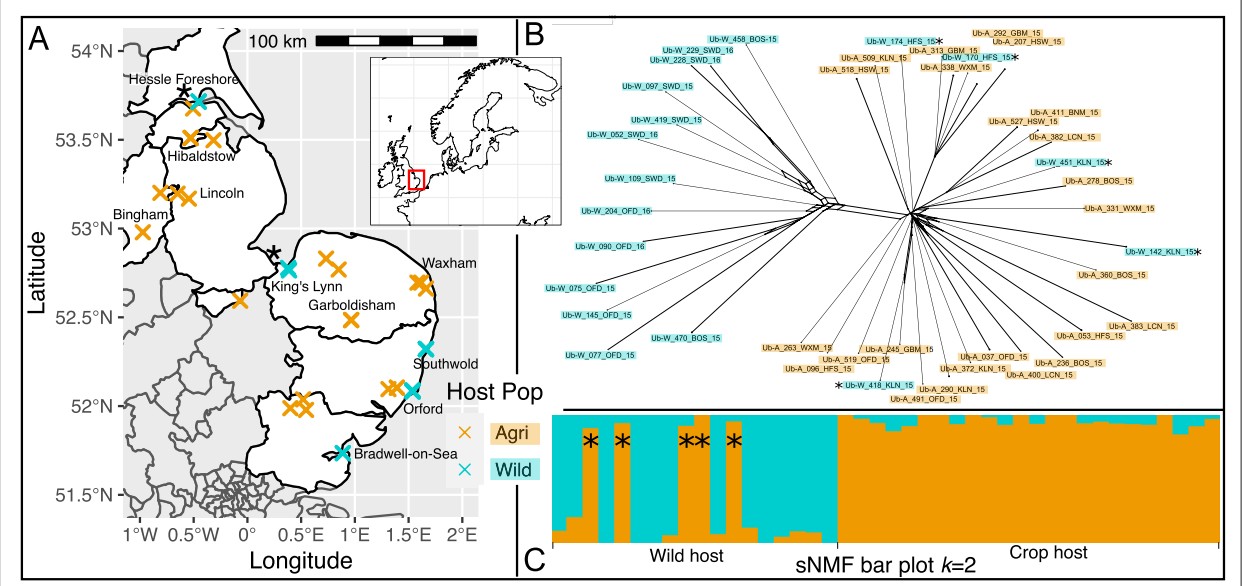

**Figure 3.** Spatial population structure and introgression of wild and crop isolates. (**A**) Map of part of the UK shows samples coloured by host (crop = cyan, and wild = orange). Overlapping crosses obscure wild samples which can be distinguished using **Supplementary file 2**. Northern wild samples are accompanied by an asterisk (continued throughout the figure) because they are found within the crop clade (see B & C, and **Figure 2**). (**B**) SplitsTree network generated using the gene coding sequence (CDS) regions present in all isolates (15.2 Mbp) shows a clear differentiation of an exclusively wild population from all crop isolates plus five wild isolates. Population clustering and admixture analysis (sparse non-negative matrix factorization, sNMF; **C**) shows two partitioned populations in which the five northern isolates sampled on wild host belong to the crop cluster, with no increased signal of hybridisation.

able to partition these five isolates away from the agricultural population without including them in the training data (Appendix 2). Combined, the evidence suggests two distinct populations (prediction A), the first of which we consider to be crop-infecting as it contains all of the crop-infecting isolates as well as five northern wild-host isolates. The second population infects exclusively wild hosts, and here we use this distinction (crop-infecting) to begin to understand pathogen evolution across a wild-crop boundary.

Network analyses were based on gene CDS regions from those genes with read coverage greater than 60% of the gene (per isolate). Low coverage genes, or gene absence did not appear to disproportionately impact one population. Low coverage impacted 87 out of 9148 genes. Most of these genes (48) were absent from all isolates. Seven genes were absent from the wild population, but they were also absent in the majority of individuals in the crop population. There was a high correlation between the presence of a gene in one population and that in another ($r^2$=0.89; p=<2.2e-16). This was the case regardless of whether isolates were grouped by DAPC clustered populations or by host affiliation (see below & Appendix 3). This pattern of gene presence/absence in effectors was the same, where four effectors were absent in all isolates.

## Effector genes provide evidence for adaptation

The sugar beet crop is not in the ground year-round, and so our expectation is that this obligate biotrophic rust invades the crop from a more diverse wild pathogen population. Natural selection from a genetically depauperate crop is predicted to disproportionally impact effectors and somewhat determine the pathogen lineage that infects the monoculture. Therefore, gene and effector genetic diversity and differentiation values respectively represent a genome-wide distribution reflecting demography (i.e. migration and drift), as well as selective forces from the hosts and the environment. We investigated differences in genetic diversity and differentiation between effector and non-effector genes to begin to unpick these processes, our null expectation being that genes would have reduced genetic diversity, and they should fall randomly around an $F_{ST}$ distribution, compared to more diverse and differentiated effectors (prediction B).

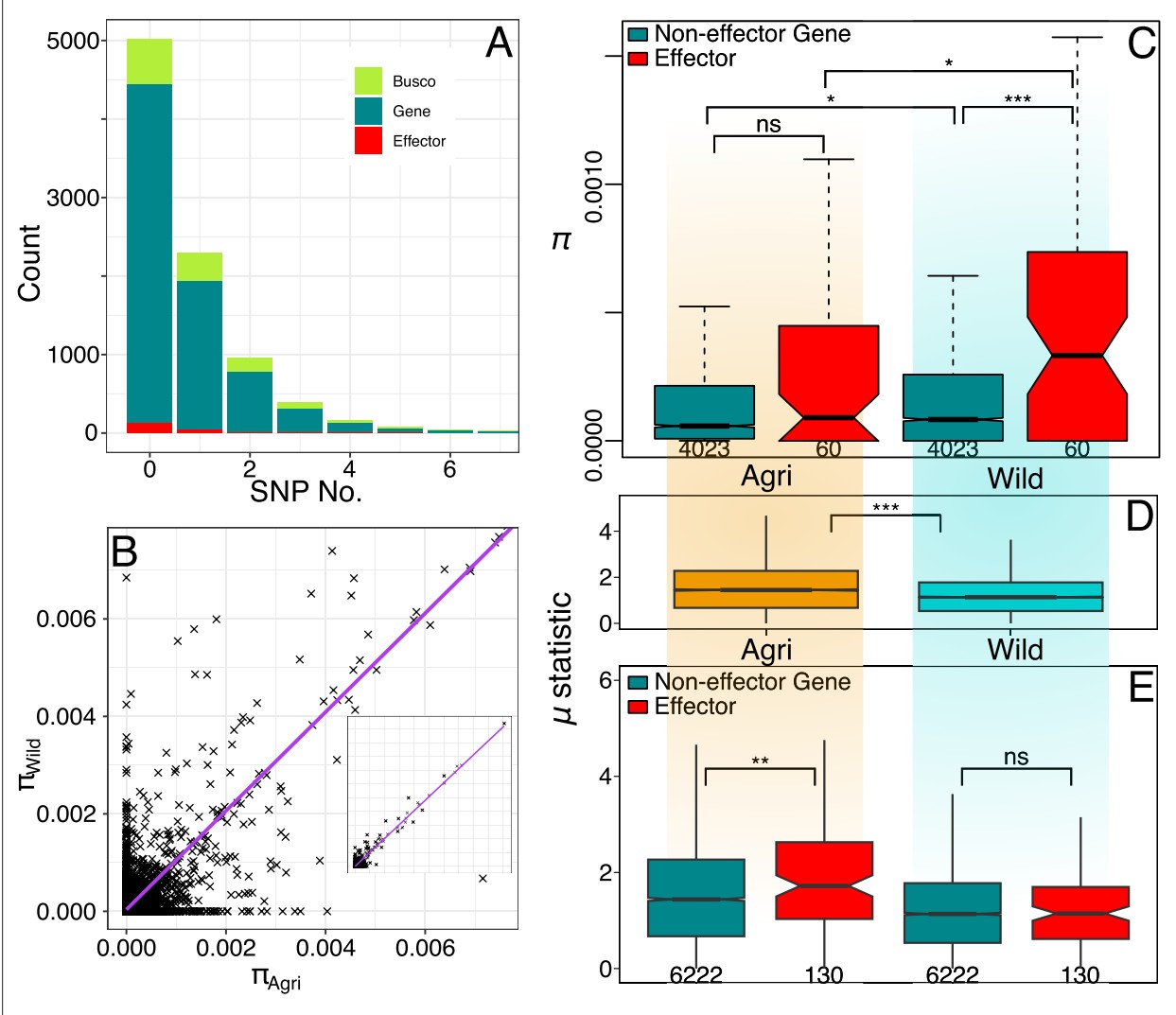

**Figure 4.** Wild effector genes maintain diversity. (**A**) Histogram of gene numbers with a given number of SNPs in the coding sequence (CDS) region (0–7 SNPs). This diversity is measured from both populations combined and shows that polymorphism is low, over half of genes don't contain a SNP within their coding region. (**B**) Nucleotide diversity present in gene CDS regions is strongly correlated between wild and crop populations. The inset shows the full range (0.00–0.05). (**C**) Nucleotide diversity present in polymorphic effector and non-effector gene CDS regions (shaded by population: Agri = orange, Wild = cyan). Agricultural non-effector genes have significantly less diversity than wild non-effector genes. Maintenance of polymorphism in wild effectors, is significantly greater than wild non-effectors and the maintenance of effector polymorphism is greater in the wild than in agriculture. The distinction between effector and non-effector genes is not significant in the agricultural population where diversity is more limited. (**D**) Raised accuracy in sweep detection (RAiSD) selective sweep $\mu$ *statistic* (50 kbp windows) plotted for all contigs (>50 kbp) from each population. The $\mu$ *statistic* is significantly greater on average in the agricultural population and the $\mu$ *statistic* plotted across each effector and non-effector gene (**E**) shows that agricultural effectors have significantly higher $\mu$ *statistic* values than non-effector genes. There is no similar significant difference in the wild population. Numbers of polymorphic genes and windows are presented under boxplots see (**Source data 1**).

At the genome scale, we found nucleotide diversity ($\pi$) in the wild reservoir population to be significantly greater than the crop population (median $\pi$ 10 Kbp windows: Agri = 0.252 × 10$^{-3}$, Wild = 0.269 × 10$^{-3}$ (93.7%); Wilcoxon($\pi$) W=2440369930, p<0.001). At the gene level, diversity was low and less than half of all gene CDS regions contained a SNP (44.6%; *Figure 4A*), while the correlation in CDS genetic diversity overall between populations is high ($r^2$=0.92, n=9093 p<0.001; *Figure 4B*). Consistent with nucleotide diversity, the level of observed heterozygosity around genes (5 kbp up- and down-stream) was greater in the wild population (median $H_O$: agri = 0.199; wild = 0.215; Wilcoxon($H_O$): W=35835730, p<0.001) but not significantly different between genes and effectors (median

$H_O$: agri-gene=0.199, agri-effector=0.195, wild-gene=0.215, wild-effector=0.208; Wilcoxon($H_O$): agri-gene-effector, W=886906, p=0.388; wild-gene-effector, W=879413, p=0.348).

We used CDS regions that were polymorphic in either population, to identify evidence for the impact of selection on effector polymorphism in reservoirs of diversity (*Figure 4C*). Consistent with genome-wide levels, genetic diversity in the wild population was higher than that of the agricultural, to the greatest extent in effectors (median $\pi$: Agri effector = 0.090 × 10$^{-3}$, Wild effector = 0.332 × 10$^{-3}$ (27.1%); Wilcoxon($\pi$) W=1445, p=0.029) as well as non-effector genes (median $\pi$: Agri non-effector=0.058 × 10$^{-3}$, Wild non-effector=0.083 × 10$^{-3}$ (69.9%); Wilcoxon($\pi$) W=7900296, p=0.031). In addition, within the wild population, effectors are more diverse than non-effectors (Wilcoxon($\pi$) W=87464, p<0.001) but in the agricultural population, this is not the case (Wilcoxon($\pi$) W=111,834 p=0.162). The median diversity of polymorphic wild effectors is more than three times greater than that of any other gene category (*Figure 4C*).

Reduced diversity at crop pathogen effectors is a prediction based on the selection of effector polymorphism by a genetically depauperate crop. Within the crop pathogen population, this may be evident as positive selection and has several expectations. In a scenario in which a polymorphism has recently been fixed, we expect a local reduction in polymorphism, local allele frequency bias, and locally high linkage (*Alachiotis and Pavlidis, 2018*). Despite the difficulties associated with using linkage-based indicators in a repetitive, fragmented genome, in addition to the demographic scenario

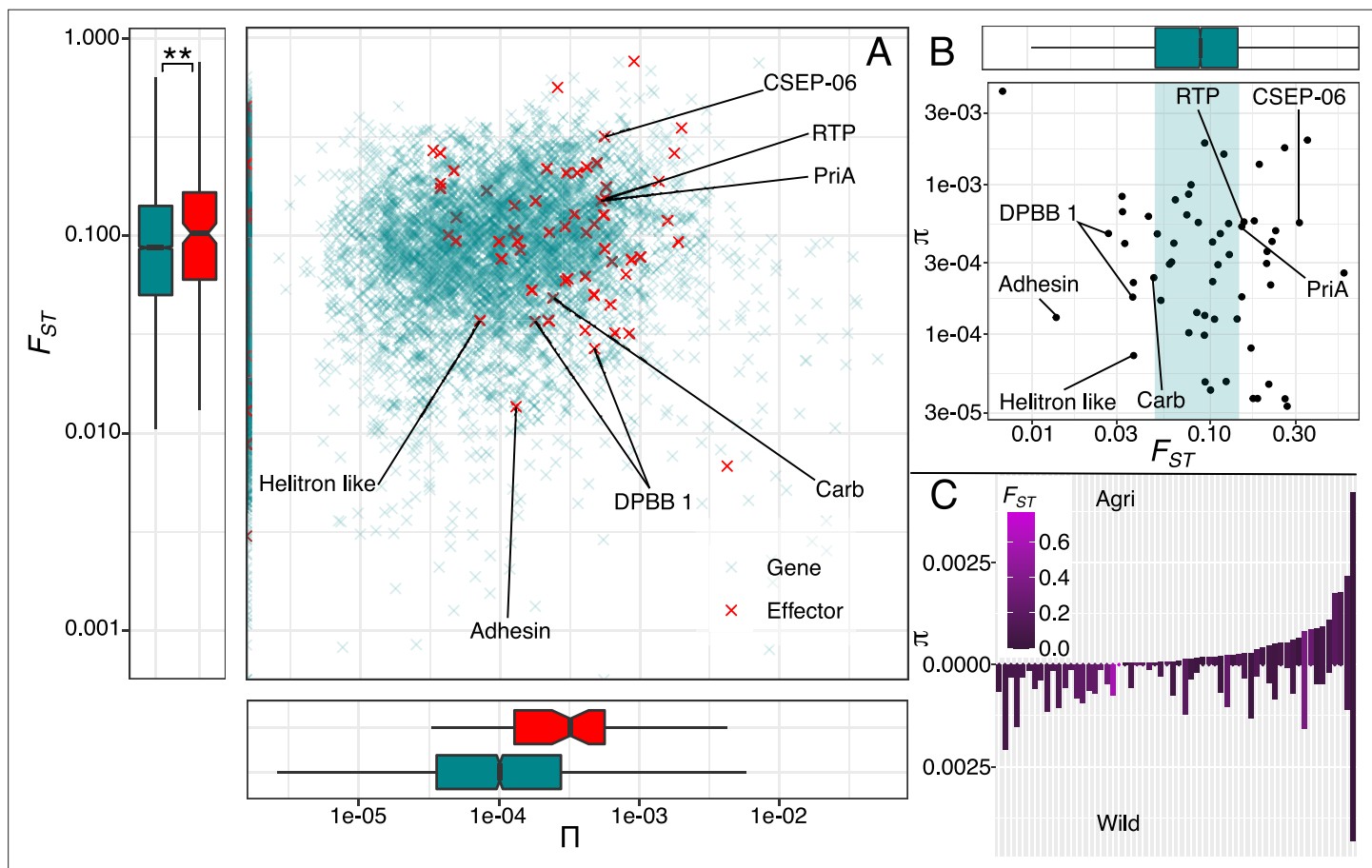

**Figure 5.** Evidence for adaptation is present in effector diversity. (**A**) Levels of nucleotide diversity plotted against genetic differentiation of effector (red) and non-effector genes (green). Genetic differentiation is significantly greater in effectors than non-effectors which suggests that more genes in this group are exposed to diversifying selection, favouring variation specific to each environment. (**B**) Polymorphic effectors are plotted against the non-effector $F_{ST}$ distribution, highlighting those effectors in the upper and lower ranges. (**C**) Polymorphic effectors plotted in order of agricultural nucleotide diversity underlines the importance of assessing agricultural diversity against a non-crop background, because this highlights those effectors, that are fixed in agriculture, are those that are most differentiated. Known effector functions in the upper and lower ranges are included (A & B, see *Source data 1*).

of invasion (also reducing polymorphism and increasing linkage; *Pavlidis and Alachiotis, 2017*), we set out to identify whether these signals were more associated with effector polymorphism.

Raised accuracy in sweep detection (RAiSD) provides a *μ statistic* which is a composite measure of the observations of a selective sweep (*Alachiotis and Pavlidis, 2018*). In general, we have avoided metrics that rely on linkage in favour of genotypic measures (diversity and heterozygosity) and so interpretations based on linkage are caveated in this large repetitive dikaryotic assembly. Importantly RAiSD is not informed by demographic simulations which, given the unknowns of the present system, could misinform. Additionally, RAiSD is known for its robustness to confounding factors. Using the *μ statistic* at 99.5%, we found zero effectors (16 non-effector genes) and only began to identify effectors at the 95.0% level, where we found 285 non-effectors and five effectors, which is not a significant overrepresentation of effectors ($\chi^2$=0.30, N=11129, p=0.58). Given the role of invasion to also reduce diversity, we assessed the relative association of the *μ statistic* in each population and found, as expected, a significant difference between the agricultural and wild populations (*Figure 4D*; median *μ*: Agri = 1.447, Wild = 1.136; Wilcoxon(π) W=23437264, p<0.001). Interestingly however, within the agricultural population, effectors are within windows of significantly greater *μ* than non-effector genes (*Figure 4E*; median *μ*: Agri-non-effector=1.439, Agri-effector=1.721; Wilcoxon(π) W=352980, p=0.006), while at the same time, there is no difference in the signal of the *μ statistic* in the wild that differentiates effector and non-effector genes (*Figure 4E*; median *μ*: Wild-non-effector=1.136, Wild-effector=1.145; Wilcoxon(π) W=402890, p=0.941).

After demonstrating greater effector diversity in the wild reservoir population, as well as a potential for fixation of effector diversity in the crop-infecting population, we next tested whether effector diversity was partitioned to a greater degree between wild and crop populations than other genes on average. The overall level of genetic differentiation ($F_{ST}$) between the two populations was calculated at $F_{ST}$ = 0.113. We predicted that depauperate host genetic diversity is one of the largest discriminating factors between the wild and agricultural environment and, therefore, predicted crop selection operating to a greater extent in the genes that interact with the host, the putative effectors. Here, we measured rust genetic differentiation at all genes (±5 Kbp flanking region) and found that effectors were indeed significantly more differentiated than non-effector genes (*Figure 5*; median $F_{ST}$: non-effector genes = 0.085, effectors = 0.103, Wilcoxon ($F_{ST}$) W=751402, p=0.001). Two of the top ten most differentiated genes are effectors, a significant overrepresentation of the effector category ($\chi^2$ (1, N=4083)=23.8, p<0.001). This signal of increased genetic differentiation at effectors was also observed from an absolute measure of divergence ($D_{XY}$; see Appendix 4).

Adaptation of virulence can also be mediated via effector silencing (*Jeon et al., 2022*). Unfortunately, this nuance is not adequately captured without expression data and, in this genome and resequencing data, we observe neither a signal of increased alternative splicing nor other high-impact nonsense mutations that could impact the functionality of the protein product (see Appendix 4).

## How might diversity statistics be used to identify important candidates?

Effectors receive fewer functional annotations than non-effector genes, which is likely a consequence of the evolutionary pressure to avoid host recognition, causing rapid diversification to reduce sequence similarity and the prevalence of conserved domains and homologs (*Franceschetti et al., 2017*). However, functional annotations were associated with 134 shared effectors that have zero polymorphism across both populations and included, among others: Rust Transferred Protein (*Pretsch et al., 2013*), conserved across the Pucciniales *Lorrain et al., 2019*; Alpha-amylase, important for degradation of plant cell walls; CSEP-06 (Candidate Secreted Effector Protein), important in Asian soybean rust *Elmore et al., 2020*; and Glycine-serine-rich candidate effector (important in Stripe rust, *Liu et al., 2022*), a Sod Cu domain-containing protein, and a thioredoxin domain-containing protein, all of which are important for suppression of host induction of reactive oxygen species (ROS), a core host response for initiation of immunity and host cell death (see functions in *Source data 1*). Of those ubiquitous effectors that were found to maintain polymorphism (across both populations), we also identified functions associated with carbohydrate-binding – carbohydrate-active enzymes (CAZymes; UniProt: A0A165X6D7 *Suzek et al., 2007*) and DPBB 1, important for degradation of polysaccharides (e.g. cellulose) in plant cell walls (IPR009009 *Blum et al., 2021*; CATH 2.40.40.10 *Sillitoe et al., 2021*).

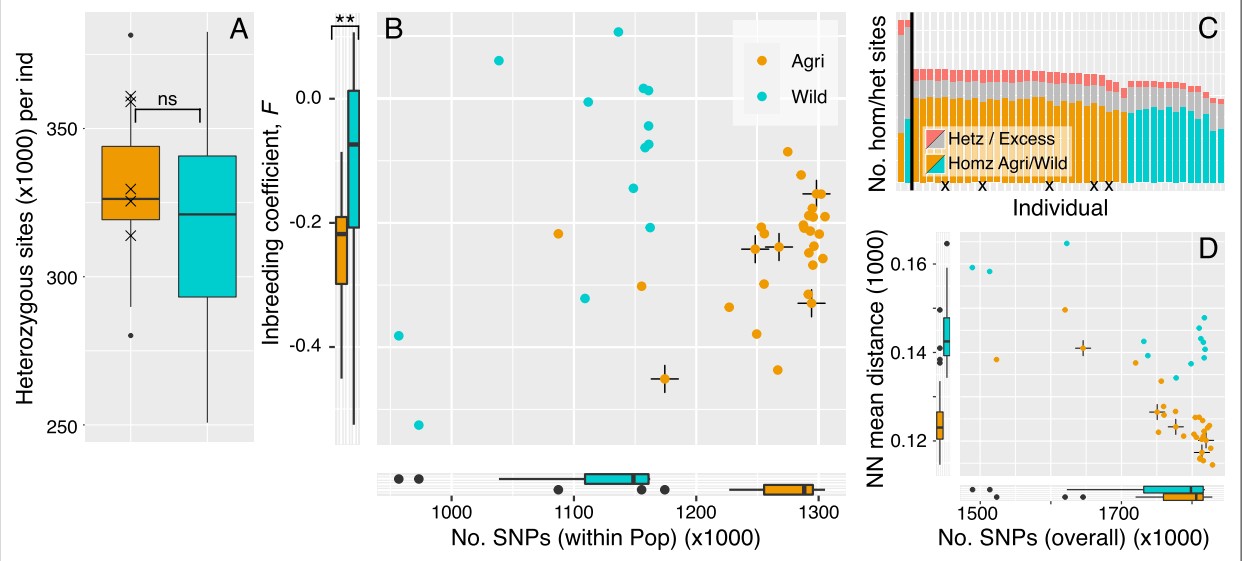

**Figure 6.** Clonality may be prioritised in the crop lineages. (**A**) The level of heterozygosity is not significantly different between crop and wild populations. Isolates that cluster within the crop population but were isolated on a wild host, are marked by cross in each panel. (**B**) Numbers of SNPs per isolate (within population), plotted against inbreeding coefficient ($F_{IS}$). Wild rust isolates have fewer SNPs segregating within population and broadly distributed $F_{IS}$ values that tend more towards zero and above. Crop isolates have larger numbers of SNPs (within population) and $F_{IS}$ values that are significantly more negative on average. Negative $F_{IS}$ values are indicative of a heterozygous excess and consistent with clonal modes of reproduction, as are accumulation of mutations at the isolate level. (**C**) Numbers of homozygous and heterozygous SNPs per isolate (across all isolates). Each isolate is represented by a single bar where homozygous sites coloured by population (Agri = orange, Wild = cyan). Grey with red tips represents the number of heterozygous sites per isolate with the red tips indicating the proportion of those heterozygous sites that are in regions of excess heterozygosity. The vertical black line separates population total values (left, reaching 1.87 million SNPs) from isolate values (right). (**D**) Numbers of SNPs per isolate, here SNPs are in relation to all other isolates plotted against mean pairwise Neighbour-Net distance among all isolates, compared to panel B.

We were particularly interested in the functions of the differentiated effectors but, unsurprisingly, there are even fewer known functions within this set (*Figure 5B*). PriA, important for priming DNA replication (*Zavitz and Marians, 1991*), has been associated with replication restart in response to DNA oxidative damage (*Feng et al., 2011*). The two most differentiated effectors in this study were also fixed in the crop population (*Figure 5C*). These differentiated and fixed genes are key to our understanding of selection within this population, which may be periodically bottlenecked by cropping. Unfortunately, we were not able to assign functions to these two highly differentiated fixed effectors *Source data 1* and they remain important candidates for further investigation.

## Reproduction may be partitioned between populations

Annual availability of the sugar beet crop may select specific pathogen genotypes from wild reservoir populations, but subsequent competition among pathogen genotypes on replicated crop hosts is expected to favour clonal reproduction, both because it is rapid but also because it preserves successful genotypic combinations (*Ali et al., 2010*). We next assessed the evidence for prediction C: that crop selection increases the rate of clonal to sexual reproduction in the population containing those isolates. Perhaps surprisingly, the level of heterozygosity is not significantly higher in the crop population, as might be expected if clonality were operating long-term (*Figure 6A*; median heterozygous site N°.: Agri = 326,246,, Wild = 321,031; Wilcoxon W=225, p=0.332; e.g. *Schwessinger et al., 2020*). To understand the relative contribution of sexual and asexual reproduction in wild and crop populations, we assessed the level of inbreeding. The inbreeding coefficient, *F*, is a genotype-based metric that describes the proportion of genetic variation contained within an individual isolate relative to its subpopulation. In most cases, *F* scales between zero and one, indicating a continuum between random mating and complete inbreeding. We found that $F_{IS}$ is negative in all isolates of the crop population (*Figure 6B*). Negative $F_{IS}$ indicates excess heterozygosity and a role for the preservation of polymorphism via clonal reproduction (*Balloux et al., 2003*). The wild population also contains isolates with negative $F_{IS}$ values, but others are positive, and this population has significantly higher $F_{IS}$

on average (*Figure 6B*; Median $F_{IS}$: Agri = −0.218, Wild = −0.074; Wilcoxon W=92, p-value = 0.008). This difference in $F_{IS}$ between wild and crop populations is consistent with a difference in the relative rates of sexual reproduction between these populations; clonal reproduction predominantly operates in agriculture, whereas reproduction is both clonal and sexual in the wild. Linkage disequilibrium is also informative for the rate of sexual reproduction (*Nieuwenhuis and James, 2016*). Given the fragmented, repeat-rich nature of the present genome, we prefer to use these genotypic measures of inbreeding and clonality. However, linkage-based analysis of recombination supported the opposite case, that recombination ($4N_er$) was indeed preserved to a greater degree in the agricultural population (Appendix 4). High contiguity phased assemblies are an important next step to resolve and discriminate the genotypic- and haplotypic-based signals.

To specifically investigate whether the heterozygosity associated with the inbreeding coefficient could have been caused by other processes, we looked for an association with other features of the genome, as excess heterozygosity could also be caused by repeat content and erroneous heterozygosity via read mismapping. Blocks of excess heterozygosity (consecutive sites of excess heterozygosity based on a 5% significance level using the Hardy-Weinberg equilibrium test) impact crop and wild populations to different extents. We found that the crop population has approximately 22 thousand blocks (two or more consecutive excess heterozygous sites) where the wild population has less than ten thousand blocks (mean block length: Agri = 227.7 bp, Wild = 124.0 bp; mean N°. excess heterozygous sites: Agri = 5.1, Wild = 5.9). We found no association between these blocks of excess heterozygosity with repeat or genic regions (Fisher's exact test for all cases: Agri-interspersed repeats, Agri-all repeats, Agri-protein-coding genes, Wild-interspersed repeats, Wild-all repeats, Wild-protein-coding genes; p=1). Therefore, we do not believe genotypic estimates for differential levels of inbreeding are caused by read mismapping.

## Host versus population grouping of rust isolates highlights non-effector genes

The analyses described thus far were based on population diversity evidence that placed five wild isolates within an agricultural clade. There are several arguments that might account for this observation. First, there is no reason to expect that a crop-host isolate lineage can never infect a wild host, particularly given the prevalence of these isolates and the expectation that this crop population may have been seeded from a wild isolate. Unfortunately, host sequencing coverage was too low to identify any relationship between host and pathogen genetic diversity (*Appendix 1—table 1*). We did additional analysis of pathogen data under a strict *crop-host* and *wild-host* criterion. In essence, this involved moving the five wild-host isolates found in the crop-host population, into the wild-host population, introducing nucleotide diversity. By forcing this direct split between wild-host and crop-host isolates we were interested to see whether there were effectors that increased levels of differentiation. There was a strong positive correlation between differentiation at all genes using both partitioning regimes and, as expected, the population partition had higher levels of differentiation (Appendix 5). This strong correlation preserved the signals observed in the previous (DAPC-based) analysis, albeit to a lesser degree. Effectors were also no more differentiated among host-partitioned isolates than for population partitioning.

## Discussion

Here, we explored the potential for population genetic measures to describe the processes underlying contemporary crop pathogen adaptation and invasion. In doing so, we set out to highlight the importance of sampling and analysing a genetic reservoir outside of the agricultural system as a means of identifying genes important for survival within that agricultural system. This wild reservoir population's genetic diversity would not have been visible had we only sampled the crop. Agriculture is a recent phenomenon and all plant pathogens, until crop domestication, evolved on wild relative hosts (*Weisberg et al., 2021*). Crop pathogens must therefore specialise, and we reasoned that the contemporary signal of this will be present in their genomes as: (A) population structure partitioned by wild and crop hosts, (B) polymorphism favoured at host interaction (effector) genes, in addition to (C) skewed reproduction towards clonality on their monoculture crop hosts (versus wild hosts). Analyses

were selected because they can be applied to any gene or fragmented genome and have applications towards pathogen genomic surveillance (*Peers et al., 2024*).

## Beet rust is differentiated into two populations in the UK

We found evidence for (prediction A) two populations overall – the population containing all the crop isolates spanned the 200 km sugar beet cropping area and surrounded the second, wild coastal population. Crop isolates were sampled just 10 km from these wild isolates at their nearest point. It is perhaps not surprising that all crop isolates belonged to the same population. However, as well as living on all crop hosts, this population continued outside of the beet cultivation area, northwards on sea beets, to the edge of our sampling range. Therefore, this crop population also included five northern wild isolates (24 crops and five wild isolates). Population genetic assignment, using PCA, DAPC, network, admixture, and machine learning analyses all identified these five northern sea beet isolates within this crop population, finding that they were no more admixed than crop isolates and did not occupy their own subclade of the network. This finding is perhaps consistent with the expectation that crop populations invade from a wild source (as well as reinfect). Without broader sampling into the northern wild range, or temporal sampling, we cannot speculate on the broader metapopulation dynamics of the source of the crop lineage and its turnover. However, host diversity is likely an important determinant of pathogen diversity and preliminary evidence from wild beet inoculation trials shows that northern (Humber) hosts represent both a distinct genetic group and have reduced resistance to crop rusts (Yvanne, et al., in prep).

To address whether a strict definition of a pathogen's host affiliation impacted our results, we also analysed diversity under that definition, ignoring population structure. As expected, this reduced the overall level of differentiation among populations but the signal in effectors remained (see below).

## Genetic reservoirs of effectors and their selection

Effectors had greater nucleotide diversity than non-effectors in the wild population but not in the crop-infecting population, which is consistent with the maintenance of polymorphism by balancing selection in the wild (*Hughes and Yeager, 1998*). This, and the observation that effectors were significantly more differentiated between wild and crop-infecting populations on average, are consistent with prediction B and suggests diversifying selection (host resistance) operating to distinguish diversity between populations. This is interesting because we do not discount the potential for effector diversity selected for both populations, reducing differentiation (*Muirhead, 2001*; *Schierup et al., 2000*). We indeed expect that this is also the case for a given subset of effectors, which makes our finding more remarkable.

Resistance gene durability, rapid breakdown, and effector evolution is an area of considerable interest (*Brown, 2015*; *McCann, 2020*). Genetic diversity and differentiation of wild pathogens is not the way to identify specific resistance-virulence gene interactions. However, the molecular analyses required to do so are time-consuming and describe interactions based on a fixed background. With real world polymorphism, these interactions may break down where they operate redundantly, epistatically, or have a dosage component (*Guo et al., 2019*; *Meile et al., 2023*; *Thordal-Christensen et al., 2018*). Human microbial genomic surveillance can already be used to identify pathogenic taxa, including antimicrobial resistance gene profiles within as little as 1 hr of sequencing (*Leggett et al., 2020*). *Melampsora lini* spatio-temporal analyses are perhaps the best analogue of the present system (*Susi et al., 2020*). In that work, the authors highlight both a role for host diversity driving pathogen selection as well as pathogen background genetic diversity originating from multiple genetic sources. Analysing pathogen diversity in this way can aid the classification of those genes most important for success on crops. For example, in the present study, two out of the top ten most differentiated genes are effectors, which is dramatically above the expected number given just 60 polymorphic effectors were identified. Moreover, these analyses broaden the focus by highlighting non-effector genes that could also increase the success of crop pathogens.

Genetic differentiation could be one of several criteria applied to define the importance of genes that are particularly successful on crop hosts. The absence of a barrier to gene flow between wild-host and crop-host isolates also provides a source of novel effector polymorphism (*McCann, 2020*) and so the level of genetic differentiation at both extremes can inform on the importance of diversity as well as sources of introduction. Effectors in general tend to receive fewer functional annotations

and, interestingly, differentiated effectors identified in the present study have fewer known functions compared to those pervasive effectors that are preserved in both wild and crop populations, and also tend to be retained among species. The present work provides a framework to identify agriculturally adapted gene candidates and subsequently identify the mechanisms that facilitate that adaptation via introgression and recombination.

## Partitioning modes of reproduction

Rusts are among those fungi able to partition sexual (recombination) and clonal life stages (*Figueroa et al., 2020*), and this strategy makes them particularly suited to invasion (*Gladieux et al., 2015a*). The beet rust lifecycle is autoecious (*Kristoffersen et al., 2018*) but the advantages of clonality (i.e. preservation of successful combinations of effectors and rapid spread on temporally available crop hosts), are as present as the advantages of recombining effectors on pervasive sexual wild hosts. Our third prediction (C), therefore, was that despite potentially high levels of gene flow (or frequent invasion), the advantages of clonal reproduction on crops would impose enough of a constraint to allow us to observe an increase in the signal of clonality in our crop-host pathogen population. Our observations here were mixed. At the genotypic level, we used isolate heterozygosity to observe a signal of clonality across the crop population. Incidentally, our five northern wild isolates from this crop population also had this signal of elevated inbreeding. Importantly, isolates from the wild reservoir population also carried signals of inbreeding alongside outbreeding, although significantly higher levels of outbreeding overall. These regions of heterozygous excess were not associated with repeats, which is consistent with an increase in the rate of clonality in agriculture, where clonal lineages preserve their heterozygosity (*Balloux et al., 2003*). At the haplotypic level, however, the estimated rate of recombination was greater for the crop-infecting population compared to the wild.

The observation that crop populations have greater excess heterozygosity, while at the same time the rate of recombination ($4N_e r$) is also higher, is unexpected for established populations. However, seasonal invasion and expansion of a crop population would preserve mutations despite frequent recombination occurring alongside the expansion, or prior to it, with its signal preserved in clonal haplotypes. So, while both analyses do confirm differentiated rates of sex in wild and crop populations, they are in opposite directions. It's important to remember that one of the reasons these systems are poorly understood, is that they are dynamic, and degrees of clonality and recombination (varied in each population) may be interacting with invasion and frequent changes in population size. These phenomena break many of the assumptions made by classical population genetics models. Unfortunately, we can't resolve all these questions here, but we highlight that more work is needed to understand pathogen invasion and adaptation to crops. Further work on single isolate, contiguous, phased genome assemblies (wild and crop) is required to add weight to haplotype-based analyses and facilitate hypothesis generation using simulations.

Investment in clonal or sexual reproduction can vary with environmental heterogeneity, where the advantages of sex for rapid adaptation diminish as environmental heterogeneity declines (*Becks and Agrawal, 2010*). Clonal pathogens spread rapidly and preserve their genotype, and this is of particular advantage on a monoculture (*Maynard Smith et al., 1993*; *Möller and Stukenbrock, 2017*). These crop pathogens do generate novel polymorphism by mutation, but without recombination, selection at one locus will interact with another and reduce the rate of adaptation (Hill-Robertson effect, *Hill and Robertson, 1966*). Rapid spread on a monoculture is perhaps a worthwhile trade-off, at least until the genetic diversity of that monoculture is improved or increased. Wild pathogens, however, must infect a new genotype with every host and so recombination is not expected to be selected against to the same extent.

Crop pathogens represent a rare opportunity to study the trade-off of clonality and sex. Wheat stripe rust has bouts of clonality spanning decades where increased levels of isolate heterozygosity is believed to be driven by lack of a local sexual host (*Schwessinger et al., 2020*). The long term impact being that the incidence of clonal wheat rust epidemics appears related to the proximity of its sexual host (*Mojerlou et al., 2025*). *M. lini* also shows temporally varying levels of clonality associated with epidemic spread on flax (*Linum marginale*) (*Susi et al., 2020*). *Balloux et al., 2003* state that mixed clonal and sexual reproductive modes are nearly indistinguishable from strict sexual reproduction. The fact that we observe excess heterozygosity in most isolates, with the wild population being impacted to a lesser extent, suggests that sexual reproduction may be infrequent in the wild and perhaps

removed from agriculture by clonal spread post-invasion (*Maynard Smith et al., 1993*). This is despite polymorphism being low (e.g. 44.6% polymorphic gene CDS), perhaps suggesting temporally fluctuating population size. Sex and the rate of clonality, inflation of effective population size, and gene flow with boom and bust dynamics make accurate modelling of the population difficult (e.g. *Susi et al., 2020*). The future impact of climate change on the availability of novel environments and environmental heterogeneity further complicates our ability to protect crops. A better understanding of genotypic- and linkage-based factors are important for this beet-rust system, but must be combined with work on the impact of lifestyle on invasion dynamics (*Taliadoros et al., 2025*, *in revision*) to allow analysis of crop pathogen genetic diversity with reservoir populations in mind. Estimation of linkage and the rate of sex is not only critical to our ability to determine where novel pathogen lineages come from, but how they continue to adapt and overcome novel sources of host resistance and treatment.

## Genomic tools for population genetics

Identifying pathogen reservoirs of diversity is arguably the first step in determining sources of introgression (or recombination). Single isolate, high contiguity phased assemblies allow differentiation and divergence metrics to be plotted at the genome level, but gene level analyses as applied here provide excellent insight from reduced resources, in the sense of sampling, sequencing, and/or assembly (*Peers et al., 2024*). To explore crop-wild plant pathogen diversity we generated the first assembly and annotation of *U. beticola* and developed a method to extract pathogen DNA for population analyses which reduces host DNA contamination. The fungal peel-sequencing method was originally developed for microscopy and was repurposed here to allow us to paint on, peel off, and genome sequence rust pustules from the surface of a leaf. This peel sequencing method was used for re-sequencing and population genetics and approximately doubled the number of isolates we could sequence.

The ~600 Mbp *U. beticola* genome is large for a fungus, but not for a rust (*Loehrer et al., 2014*). The rusts have dikaryotic haploid nuclei and in many rusts, this produces a signal (Meselson effect) in which extended periods of clonal reproduction increase the divergence of homologous content (addressed from a population context above, see also, *Schwessinger et al., 2020*; *Mark Welch and Meselson, 2000*). For population analyses, the number of isolates analysed here is relatively small. This perhaps makes the signals we observe even more striking. Further research in this system could go in each of two directions. Improving the assembly would facilitate stronger linkage analyses, more accurate estimation of recombination rate and then modelling invasion and population divergence scenarios. Targeted approaches would facilitate collection and analysis of rust pathogen populations in a new season at multiple locations across Europe. By not paying the price of whole genome sequencing for each isolate, analyses could focus on identifying evidence for selection of the same alleles on crop hosts in multiple independent populations across Europe (*Peers et al., 2024*).

Pathogen genetic reservoirs may be operating in a number of crop wild relatives, or alternative hosts. Many agronomically important rusts have an alternate host which is used by the fungus to enter its sexual phase (*Aime et al., 2017*). In the case of the wheat rusts, hundreds of years before a causative link was established, superstition drove the removal of its alternative host, *Berberis*, from cereal-growing regions (*Barnes et al., 2020*). *U. beticola* reproduces on wild sea beet and, the permanent proximate availability of this host offers an explanation as to why we have not observed such divergence of karyon in *U. beticola*.

## Conclusions

Once we acknowledge that crop pathogens exchange genetic variation and even invade from (non-crop) genetic reservoirs, we underline the importance of incorporating these ideas into our understanding of how crop pathogens overcome host resistance. We set out to explore the utility of analysing genetic diversity of crop pathogens against a non-crop (wild) background, to highlight crop isolate polymorphism which would not otherwise have stood out as important for adaptation and emergence. In this first analysis of the beet-rust system we found, as expected, that effectors were disproportionately impacted by predicted host selection. In line with aims to reduce the blanket use of fungicides (*Kristoffersen et al., 2018*; *Langdale, 2021*), identification of potential sources of pathogen emergence will allow more targeted treatment of those agricultural areas deemed important for invasion. Further investigation of this system will shed light on these areas, but analysis

of multiple systems using a crop-reservoir framework will improve our understanding of pathogen evolution. This broader knowledge has implications for crop protection via pathogen surveillance, virulence gene identification, and emergence prediction.

## Materials and methods

### Genome assembly and annotation of *U. beticola*

A rustinfected sugar beet plant from Norfolk was placed in a Snijders growth cabinet (16 hr day at 16°C and 14°C at night) to allow the infection to progress in the absence of agitation by wind and rain. Multiple pustules (*Figure 1*) from a single heavily infected leaf (sampled in Garboldisham, *Figure 3A*) were broken and spores collected for DNA extraction (see below for CTAB details) for DISCOVAR PCR free library preparation and sequencing (*Weisenfeld et al., 2014*). A HiSeq 2500 (Illumina inc) was used to sequence 43.8 Gbp of data which was estimated to be 73 x coverage based on genome size estimates of ~600 Mbp generated using k-mer coverage of skim sequenced peels (*Supplementary file 3*). Post assembly, contigs less than 1 Kbp were removed as they don't represent real terms increase in the span of a single read pair. ABɣSSv2.0.2 (*Simpson et al., 2009*), KATv2.3.4 (*Mapleson et al., 2016*), BʟᴏʙTᴏᴏʟs v0.9.19 *Laetsch and Blaxter, 2017* and BUSCOv4.0 (*Simão et al., 2015*; against basidiomycota_odb10) were used to assess genome content, contiguity and completeness. BʟᴏʙTᴏᴏʟs was used to retain contigs with BLAST hits to Basidiomycetes as well as those without a hit.

To generate a genome annotation, RNA was extracted from rust-infected sugar beet leaves from multiple beets grown in the same field. Using a soft-bristled toothbrush, pustules were brushed from each leaf into a 2 ml microfuge tube with the aim of combining 100 mg of material per tube. Spores were then flash-frozen in liquid nitrogen in preparation for RNA extraction (10 tubes). In addition, leaf punches were also taken around pustules away from vascular elements of the leaf to capture fungal expression in planta (2 samples). RNA was extracted using the Qiagen AllPrep Fungal DNA/RNA/ Protein Kit and stranded libraries were prepared using the NEBNext Ultra II Directional with poly-A selection. Libraries were sequenced on one lane of the NovaSeq 6000, SP flow cell with the 300-cycle kit (150 bp PE) with v1 chemistry, Illumina inc.

Gene models were annotated using a workflow which incorporated repeat identification, RNA-Seq mapping and assembly, and alignment of protein sequences from related species (see *Supplementary file 2*). Alternative reference guided assembly methods were employed (StringTie 2: *Kovaka et al., 2019*; Scallop: *Shao and Kingsford, 2017*) to assemble transcripts for each sample. From these, a filtered set of non-redundant transcripts were derived using Mikado (*Venturini et al., 2018*). Gene models were classified based on alignment to protein sequences, identifying the subset of gene models with likely full-length ORFs. The classified models together with aligned proteins and repeat annotation are provided as hints to AUGUSTUS (*Stanke et al., 2006*). Three alternative AUGUSTUS gene builds were generated using different evidence inputs or weightings. These together with the gene models derived from the Mikado transcript selection stage were consolidated into a single set of gene models using Minos (*Kaithakottil et al., 2020*: https://github.com/EI-CoreBioinformatics/minos). The Minos pipeline scores alternative models based on the level of supporting evidence (protein homology, transcriptome data) and gene structure characteristics (e.g. CDS, UTR features) to select a representative gene model and alternative splice variants.

RNA-Seq alignment and transcript assembly (*Supplementary file 2*, tabs **Reads Alignment** and **Transcript Assemblies**) began with alignment of RNA-Seq PE reads aligned to the genome using HISAT2 v2.1.0 (*Kim et al., 2019*) (with option: `--dta --min-intronlen=20 --max-intronlen=50000 --rna-strandness RF`). The aligned reads were assembled using StringTie2 v1.3.3 (with option: `--rf`), as well as Scallop v0.10.2 (with option: --library_type first). High confidence junctions were identified with Portcullis v1.1.2 (*Mapleson, 2018*: https://github.com/EI-CoreBioinformatics/portcullis) (with options: full `--threads 8 --orientation FR --canonical C,S --min_cov 2 --save_bad --strandedness firststrand`).

Mikado was used to integrate the transcript assemblies (StringTie and Scallop) and select the best (highest scoring) model at each locus (see *Supplementary file 2*, tab **Mikado Transcript**). Mikado utilises intrinsic metrics based on ORF's predicted using prodigal v2.6.3 (*Hyatt et al., 2010*) (with options: -g 1 f gff), and extrinsic metrics derived from BLAST (v2.6.0, blastx; *Altschul et al., 1997*) searches against the cross-species reference protein database using diamond v0.9.24 (*Buchfink et al.,*

*2015*) (with options: blastx --`outfmt` 6 qseqid sseqid pident length mismatch gapopen qstart qend sstart send evalue bitscore ppos btop) and junctions passing portcullis filtering.

Mikado was used for gene predictor training (see *Supplementary file 2*, tab **Augustus Training**), where 'True' transcript models were further classified into three categories, namely:

- Gold: Models with Full-lengtherNEXT (v0.0.8; *Seoane et al., 2012*); using fungal species downloaded on 04Feb2020 hit of 'Complete/Putative Complete' and with at most two complete five_prime_UTR's and three five_prime_UTR's and at most one complete three_prime_UTR and two three_prime_UTR's

  - Silver: Models with CDS length ≥ 900 bps and with at most two complete five_prime_UTR's and three five_prime_UTR's and at most one complete three_prime_UTR and two three_prime_UTR's
  - Bronze: Models that did not classify as Mikado transcript Gold or Silver

A subset of Mikado Gold transcripts were selected for training AUGUSTUS i.e., those with a single full-length ORF, 5' and 3' UTR present, consistent Full-lengtherNEXT and CDS coordinates, a minimum CDS to transcript ratio of 50%, and a single transcript per gene. We excluded genes with a genomic overlap within 1000 bp of a second gene and gene models that are homologous to each other with a coverage and identity of 80%. The filtered Mikado Gold set contained 2412 transcripts for training AUGUSTUS. The trained AUGUSTUS model resulted in 0.963 sn, 0.915 sp nucleotide level, 0.732 sn, 0.718 sp exon level and 0.33 sn, 0.301 sp at the gene level.

Cross-species proteins informed annotation (see *Supplementary file 2*, tab **Protein Alignments**) by first soft-masking using segmasker (blast v2.6.0) and alignment to the low complexity soft-masked genome (RepeatMasker v4.0.7; *Smit et al., 2008*) Combined Database: Dfam_Consensus-20170127, RepBase-20170127 using pucciniaceae species using exonerate v2.4.0 (*Slater and Birney, 2005*) (with options: --`model protein2genome` --showtargetgff yes --`showvulgar` yes --querychunkid 10 M 281.25 -D 281.25 --hspfilter 100 --`softmaskquery yes` --softmasktarget yes --bestn 10 --`minintron 20` --maxintron 20000 --`showalignment no` --geneseed 250 percent 30 --score 50 --ryo '>%qi\tlength=%ql\talnlen=%qal\tscore=%s\tpercentage=%pi\nTarget>%ti\tlength=%tl\talnlen=%tal\n') and the protein alignments were filtered at 50% identity and 80% coverage.

The evidence guided annotation of protein-coding genes based on repeats, RNA-Seq mapping, transcript assembly, and alignment of protein sequences was created using AUGUSTUS v3.3.3 (with options: --`AUGUSTUS_CONFIG_PATH` = trained_species_config –species = Uromyces_beticola --`UTR`=on --`alternatives-from-evidence`=true --`noInFrameStop`=true --`allow_hinted_splicesites` = atac).

We used the RepeatModeler v1.0.10 (*Smit et al., 2008*), library of repeats from the Uromyces_beticola_EI_v1.1.genome.fasta genome (see *Supplementary file 2*, tab **Repeats**).

Subsequent steps were:

1. RepeatMasker v4.0.7 with RepBase Pucciniaceae library (RepBaseRepeatMaskerEdition-20170127.tar.gz)
2. RepeatMasker with the RepeatModeler library

**Table 2.** Evidence sets for AUGUSTUS run in three ways.

| Evidence | Run1 | Run2 | Run3 |
|---|---|---|---|
| Mikado Transcript Gold | Source M; Priority 10; | Source M; Priority 10; | Source E; Priority 10; |
| Mikado Transcript Silver | Source F; Priority 9; | Source F; Priority 9; | Source E; Priority 9; |
| Mikado Transcript Bronze | Source E; Priority 8; | Source E; Priority 8; | Source E; Priority 8; |
| Mikado Transcripts | Source E; Priority 7; | Source E; Priority 7; | Source E; Priority 7; |
| Portcullis Pass Gold (score = 1) | Source E; Priority 6; | Source E; Priority 6; | Source E; Priority 6; |
| Portcullis Pass Silver (score <1) | Source E; Priority 4; | Source E; Priority 4; | Source E; Priority 4; |
| Proteins | Source P; Priority 4; | Source P; Priority 4; | Source P; Priority 9; |
| RNA-Seq Coverage Wig Hints | Source W; Priority 3; | NONE | NONE |
| Repeats | Source RM; Priority 1; | Source RM; Priority 1; | Source RM; Priority 1; |

The above interspersed repeats were combined and used for the gene build.

AUGUSTUS was run three different ways by assigning higher bonus scores and priority based on evidence type and classification (Gold, Silver, Bronze) to reflect the reliability of different evidence sets (*Table 2*):

Run1: Run utilizes the evidence hints generated from Mikado transcript models, RNA-Seq Portcullis junctions, cross-species protein alignments (filtered at 80% coverage and 50% identity), RNA-Seq read coverage, and interspersed repeats by using the sources and priorities (*Table 2*).

Run2: Run uses same evidence hints as Run1, except that we do not use the RNA-Seq read coverage hints.

Run3: Run uses same evidence hints as Run2, except that we give higher weightage to the cross-species protein alignments, and we also change the priorities (*Table 2*).

The final set of gene models was selected using Minos (see *Supplementary file 2*, tab **Minos Release**). Minos is a pipeline that generates and utilises metrics derived from protein, transcript, and expression data sets to consolidate gene models. In this annotation, the three alternative Augustus gene builds described earlier, and the gene models derived from the Mikado transcript run were consolidated into a single set of gene models.

Assignment of gene biotypes and confidence classification of gene models as biotypes 'protein-coding gene,' 'predicted gene,' and 'transposable element gene', and assigned as high or low confidence based on below criteria:

### High confidence protein-coding gene
Any protein-coding gene where any of its associated gene models have a BUSCO v4.0.6 (*Seppey et al., 2019*) protein status of Complete/Duplicated OR have blastp (v2.9.0+) coverage (average across query and target coverage) ≥ 80% against the list protein datasets mentioned in Section 4. Or alternatively have average blastp coverage (across query and target coverage) ≥ 60% against the list protein datasets AND have transcript alignment F1 score (average across nucleotide, exon, and junction F1 scores based on RNA-Seq transcript assemblies) ≥ 40%.

### Low confidence protein-coding gene
Any protein-coding gene where all of its associated transcript models do not meet the criteria to be considered as high confidence protein-coding transcripts.

### High confidence transposable element gene
Any protein-coding gene where any of its associated gene models have coverage ≥ 40% against the combined interspersed repeats mentioned in Section 5. Gene build/Repeats section.

### Low confidence transposable element gene
Any protein-coding gene where all of its associated transcript models do not meet the criteria to be considered as high confidence and assigned as a transposable element gene (see c).

### Low confidence predicted gene
Any protein-coding gene where all of its associated transcript models do not meet the criteria to be considered as high confidence protein-coding transcripts. And, in addition where any of the associated gene models have average blastp coverage (across query and target coverage)<30% against the list protein datasets mentioned in Section 4 AND having a protein-coding potential score <0.25 calculated using CPC2 0.1 (*Kang et al., 2017*).

### Discarded models
Any models having no BUSCO protein hit AND no protein alignment score (average across nucleotide, exon, and junction F1 scores based on protein alignments) AND no transcript alignment F1 score (average across nucleotide, exon, and junction F1 scores based on RNA-Seq transcript assemblies) AND no blastp coverage (average across query and target coverage) AND Kallisto v0.44 (*Bray et al., 2016*) expression score <0.3 from across RNA-Seq reads OR having short CDS <30 bps.

All proteins were functionally annotated using AHRD v.3.3.3 (*Hallab et al., 2017*: https://github.com/groupschoof/AHRD). Sequences were blasted against the UniProt fungi sequences (data download date 04Jun2020), both Swiss-Prot and TrEMBL datasets (*Apweiler et al., 2004*). Proteins were BLASTed (v2.6.0; blastp) with an e-value of 1e-5. We have also provided InterProScan (v5.22.61; *Jones et al., 2014*) results to AHRD. We adapted the standard AHRD example configuration file path test/resources/ahrd_example_input_go_prediction.yml, distributed with the AHRD tool, changing the following apart from the location of input and output files:

1. we included the GOA mapping from uniprot (ftp://ftp.ebi.ac.uk/pub/databases/GO/goa/UNIPROT/goa_uniprot_all.gaf.gz) as parameter 'gene_ontology_result'
2. we also included the interpro database (ftp://ftp.ebi.ac.uk/pub/databases/interpro/61.0/interpro.xml.gz) and provided as parameter 'interpro_database,'
3. we changed the parameter 'prefer_reference_with_go_annos' to 'false' and did not use the parameter 'gene_ontology_result,'
4. we have only used swissprot and trembl as blast_dbs databases

In plant pathogenic fungi, gene products that are secreted outside of the fungal cell and into the host are considered candidate host interaction genes, potentially facilitating infection. These putative effectors are identified here using signal peptide information, genes with the presence of a signal peptide using SIGNALP v3.0: (-s notm -u 0.34; *Bendtsen et al., 2004*) and the absence of transmembrane or mitochondrial localization signals using TMHMM v2.0 and TARGETP-2.0 Server (*Emanuelsson et al., 2007*) were finally assessed for sequence similarity to known effectors using EFFECTORP2.0 (*Sperschneider et al., 2018*). These putative effectors (henceforth, effector) were not functionally validated in the present study. However, we did use signals of diversity and differentiation to begin to categorise effectors into those that were more differentiated. This was initially done for those polymorphic effectors in the five percent most extreme levels of the differentiation scale, however, because effectors in general receive fewer functional annotations in the databases, none of those effectors had a functional annotation (*Supplementary file 2* and *Source data 1*). Therefore, we broadened this criterion to those in the top twenty-five percent of the non-effector gene distribution. Future work to formally characterise differences in pervasive effectors from crop-adapted ones will shed light on the roles of these genes.

## *U. beticola* isolate collection and sequencing

To identify wild sample sites, in 2015 we explored coastal sites on the east of England where sea beet was known to grow. To help locate these sites we used a number of online records of beta maritima distributions as a guide, such as the Botanical Society of Britain and Ireland species distribution list (https://database.bsbi.org/maps/), the Online Atlas of the British and Irish Fauna (https://plantatlas.brc.ac.uk/) and the National Biodiversity Network Atlas (https://nbnatlas.org/). Between the months of September 2015 to December 2016, samples of numerous beet folia pathogens (546) were collected as leaf peels (see below). Wild sites of approximately 5 km were surveyed to identify beets and their infections and wild sea beet rust samples (81) were collected from UK east coast sites between Southminster and Hull (~370 km of coastline). Sugar beet rust samples (326) were collected by the British Beet Research Organisation (BBRO) by beet growers covering approximately the same latitudes (~200 km). For wild sites, we sampled multiple plants at different patches per site, that ranged in size from a few hundred meters to approximately 5 km (*Supplementary file 2*).

DNA extraction from single rust pustules was performed at the Earlham Institute (EI) using the peel extraction and sequencing method. This method involves pipetting 5 µl cellulose acetate onto a rust pustule and allowing the solution to air dry for 1 min before peeling the fungus away from the leaf (*Figure 1C and D*). Peels were stored (−80 °C) prior to DNA extractions by agitation and modified phenol-chloroform methodology. This peel method maximises pathogen to host DNA in the extraction and avoids excess sequencing contamination of the host genome.

Peels were ground using one 4 mm and five 1 mm stainless steel ball bearings in a TissueLyser (Qiagen, Valencia, CA) for 60 seconds at a frequency of 22 Hz. Fragmented peels were then incubated at 50 °C for 1 hr in 500 µl extraction buffer 2% cetyltrimethylammonium bromide, 1.4 M NaCl, 20 mM EDTA (pH 8), 100 mM Tris-HCl (pH 8), 0.2% β-mercaptoethanol, 1 mgml⁻¹ proteinase K (*Mathers et al., 2019*). Phenol:Chloroform:Isoamyl Alcohol 25:24:1, saturated with 10 mM Tris, pH 8.0, 1 mM EDTA (500 µl) was added to each sample and vortexed before centrifugation at 14,000 g for 5 min.

The aqueous (upper) phase (~450 µl) was harvested into a fresh microcentrifuge tube. To this 100% (v/v) of Agencourt AMPure XP (Beckman Coulter) or a homemade mix Sera-Mag Speed-beads (Fisher Scientific, cat. #65152105050250) in a PEG/NaCl buffer beads (*Rohland and Reich, 2012*) magnetic beads were added, samples were vortexed for 20 s and incubated for 10 min at room temperature. The tubes were placed on a magnetic stand, and supernatant was removed and discarded after the beads had been drawn to the magnet (~2 min). The beads were washed three times with 1 ml of 80% EtOH. EtOH was removed and the beads allowed to air dry for 5 min. Magnetic beads were then resuspended in 55 µl of EB buffer (10 mM Tris-HCl) and tubes incubated at 37 °C for 10 min (vortexing every 2 min). To eluate 1 µl of 10% (v/v) RNase A (100 mgml$^{-1}$) was added before incubating at 37 °C for 30 min. Magnetic beads (50 µl) were added to each sample, vortexed and incubated at room temperature for 10 min. Sample tubes were then placed onto a magnetic rack for 2 min and the supernatant discarded. Beads were washed twice with 200 µl of 80% EtOH. Ethanol was removed and the beads left to air dry for 5 min before resuspending in 55 µl of TLE buffer (Tris low EDTA – 10 mM Tris-HCl, 0.1 mM EDTA). The tubes were incubated at 37 °C for 10 min (vortexing every 2 min). The tubes were again placed onto a magnetic rack for 2 min and the cleared eluate was harvested and stored at –20 °C.

Libraries prepared using leaf peels can differ in levels of biological contamination (from the host and from other leaf surface microorganisms). To avoid sequencing heavily contaminated peels we screened our samples using skim sequencing. Data from two lanes of HiSeq2500 (125PE, Illumina Inc) was used to align the reads to the rust reference (Samtools view -q 60; see below). Rust samples considered for resequencing were from those 300 libraries with the greatest coverage of the reference (see *Supplementary file 3*). In total, there were 404 rust samples with a mean mapping coverage of 8.4 mb, ranging from 280 bp-18.78kbp, with the top 300 samples having a read coverage of greater than 8mbp. Libraries were prepared using the LITE method *Perez Sepulveda et al., 2020* and 46 isolates were genome sequenced at the Earlham Institute on nine lanes of an Illumina HiSeq4000 (150PE, Illumina Inc) generating ~750 Gbp of read data.

## Mapping and SNP calling

Reads were quality trimmed (minimum length 90, quality 30, `--paired`; TRIM_GALORE v0.4.0; Babraham Institute, Cambridgeshire, UK). Reads were then aligned to the reference using BWA mem (*Li, 2013*), SAMTOOLS v1.5 (*Li et al., 2009*), and BCFTOOLS v1.3.1 were then used to sort and remove duplicate reads and mpileup (-t DP) to call variants (bcftools call -c). VCFTOOLS v0.1.13 (*Danecek et al., 2011*) was used to filter SNPs to an isolate minimum depth of five, maximum depth of 1.8x mean depth per isolate, and a minimum genotype quality of 30. SNP sites with more than two alleles were excluded as probable errors. Finally, sites that were missing in 30% or more isolates were also removed. Four isolates were removed from further analysis because their average depth was less than 10 x leaving 42 isolates for population analyses (*Appendix 1—table 1*).

## SNP diversity and differentiation analysis

Population differentiation was determined using principal component analysis and discriminant analysis of principal components (Adegenet:PCA, *Jombart, 2008*; DAPC, *Jombart et al., 2010*; vcfR v1.12.0, *Knaus and Grünwald, 2017*). PCA was based on 307,041 linkage-filtered SNPs (PLINK v2.0.0, -indep 50 10 0.1; *Purcell et al., 2007*) and DAPC based on all 1.87 million SNPs. The *find.clusters* (2-20) function, with all 41 principal components, identified two genetic clusters based on the lowest Bayesian information criterion (BIC). DAPC was then run using 15 principal components, accounting for 68.7% of the conserved variance. Clone correction (*mlg.filter*) was conducted using a filtered SNP dataset using 1 SNP per 100 Kbp (1.1% of the total) and no clonal genotypes were condensed in the dataset regardless of the threshold used from *filter_stats* (farthest, average, or nearest). All 42 isolates were retained for further analysis.

The R package LEA was used to calculate individual ancestry coefficients with the sNMF function using all SNPs. The function was run using default values for 'alpha' (10), 'tolerance' (0.00001), 'percentage' (0.05), 'iterations' (200), and 'ploidy' (2). The *K* parameter, or the number of assumed subpopulations was set to 1:10. The number of repetitions for sNMF to perform for each value of *K* was set to 10.

VCFTOOLS was used for population analyses of nucleotide diversity ($\pi$), heterozygosity, Hardy-Weinberg, and $F_{ST}$. Nucleotide diversity comparisons between wild and crop populations were conducted using 10Kbp windows and in gene CDS regions. Approximately half of all gene CDS regions contained zero SNPs and so for gene level comparisons of nucleotide diversity only genes polymorphic in one or other population were used. $F_{ST}$ analyses of gene regions used 5 Kbp up- and downstream of gene boundaries. The Wilcoxon Signed Rank Tests were used to determine differences in the levels of genetic diversity and differentiation among genes. The inbreeding coefficient, $F$ was calculated to identify levels of excess heterozygosity (--het, within isolates) and observed to expected levels were compared among isolates to identify blocks of excess heterozygosity (--hardy). This method provides a test statistic per site for deviations from Hardy-Weinberg equilibrium and consecutive sites of significant excess heterozygosity (at $p \leq 0.05$) were recorded as blocks of excess heterozygosity. Associations of excess heterozygosity with genomic features (e.g. repeats and genic regions) were tested within BEDTOOLS v2.26.0 (**Quinlan and Hall, 2010**) using a Fisher's exact test.

DNASP v6.12.01 (**Rozas et al., 2003**) was used to calculate population genetic statistics per gene CDS for five groups (or populations) of isolates. These groupings included all individuals, agricultural individuals as defined by population (DAPC), wild individuals as defined by population (DAPC), crop-infecting individuals as defined by host, and wild individuals as defined by host. These analyses were conducted on phased alleles after two levels of phasing, first using SNPs present on phase informative reads and then using population genetic linkage of homozygous SNPs to infer haplotypes from population data (SHAPEIT v2.20: assemble `--states` 1000 `--burn` 60 `--prune` 60 `--main` 300 `--effective-size` 88000 `--window` 0.5, **Delaneau et al., 2013**). The effective population size estimate ($\vartheta=4Ne\mu$) for SHAPEIT was calculated using mean $\pi$ from 50 kb windows from across the genome ($\pi=0.000352393$) using an assumed mutation rate, $\mu=1 \times 10^{-9}$ (**Kasuga et al., 2002**). Ten contigs (19 genes) failed the phasing step and were removed from further analysis. Fasta conversion from vcf (GATK v3.5.0 -FastaAlternateReferenceMaker, **McKenna et al., 2010**) was performed for spliced CDS regions (Cufflinks v2.2.1 -gffread, **Trapnell et al., 2010**) per isolate for all genes with read coverage greater than 60%. Statistical analyses and plotting were performed using R v3.6.3.

CDS gene diversity data for 15,612 genes (and TEs) from DNASP alongside variant impact data (SnpEff; **Cingolani et al., 2012**) are combined into a single table with functional annotations and genetic differentiation ($F_{ST}$) for gene regions (±5 kbp) (**Source data 1**). Isolate diversity is represented using default parameters SPLITSTREE v4 (**Huson and Bryant, 2006**; genes present in all isolates, 15.2 Mbp) for 15,473 gene CDS regions present in all isolates (minimum 60% coverage, Appendix 3).

RAiSD v2.9 (**Alachiotis and Pavlidis, 2018**; Raised Accuracy in Sweep Detection) uses SNPs within a sliding window (50 SNPs) to calculate a measure of positive selection, the $\mu$ statistic, which is based on three signals of a selective sweep, including: local reduction in polymorphism, site frequency spectrum skewed towards low- and high-frequency derived variants and, increased local linkage. Important also is that RAiSD requires no other input parameters. RAiSD was used on all SNPs separately in the wild (1,172,766) and crop (1,309,150) populations. We used the $\mu$ statistic in two ways, first to identify genes within windows above given thresholds and second to determine the level of evidence for selective sweeps around genes in each of the wild and crop populations.

## Acknowledgements

We are grateful to the BBRO, their disease scouts, and ultimately to UK beet growers for sending rust-infected sugar beet leaves. We are also grateful to Wilfried Haerty and Conrad Nieduszynski of the Earlham Institute and Cock van Oosterhout of the University of East Anglia for critical comments to the manuscript. *U beticola* genome sequencing and re-sequencing was supported by the British Beet Research Organisation (BBRO) pump-priming award to MC (LPA & MM). Work at the Earlham Institute (EI) was supported by the Biotechnology and Biological Sciences Research Council (BBSRC), part of UK Research and Innovation, through the Core Capability Grant (BB/CCG1720/1) and transcriptome sequencing via (BBS/E/T/000PR9818) WP1 Signatures of Domestication and Adaptation. Sequencing at the EI is supported by BBSRC National Capability in Genomics and Single Cell Analysis (BBS/E/T/000PR9816) by members of the Genomics Pipelines and Core Bioinformatics Groups. The authors also acknowledge Research Computing at the James Hutton Institute for providing computational resources and technical support for the UK Crop Diversity Bioinformatics HPC (BBSRC grants

BB/S019669/1 and BB/X019683/1), which contributed to the results presented in this paper. HY was supported by the BBSRC-funded Norwich Research Park Biosciences Doctoral Training Partnership grants BB/T008717/1.

## Additional information

### Funding

| Funder | Grant reference number | Author |
| --- | --- | --- |
| Biotechnology and Biological Sciences Research Council | BB/CCG1720/1 | Neil Hall |
| Biotechnology and Biological Sciences Research Council | BBS/E/T/000PR9818 | Neil Hall |
| Biotechnology and Biological Sciences Research Council | BBS/E/T/000PR9816 | Neil Hall |
| Biotechnology and Biological Sciences Research Council | BB/T008717/1 | Hélène Yvanne |

The funders had no role in study design, data collection and interpretation, or the decision to submit the work for publication.

### Author contributions

Mark McMullan, Conceptualization, Data curation, Formal analysis, Supervision, Funding acquisition, Investigation, Visualization, Methodology, Writing – original draft, Project administration, Writing – review and editing; Lawrence Percival-Alwyn, Conceptualization, Data curation, Formal analysis, Supervision, Funding acquisition, Investigation, Methodology, Project administration, Writing – review and editing; Gemy G Kaithakottil, Resources, Data curation, Formal analysis; Laura-Jayne Gardiner, Software, Formal analysis, Investigation; Rowena Hill, Formal analysis, Writing – review and editing; Hélène Yvanne, Sabrina Jaye Ward, Jakob Kroboth, Formal analysis; Michelle Grey, Darren Heavens, Methodology; Kevin Sawford, Investigation, Methodology; Ross Low, Sally D Warring, Ned Peel, Formal analysis, Methodology; Mark Stevens, Supervision, Project administration; David Swarbreck, Data curation, Formal analysis, Supervision, Investigation, Methodology, Project administration; Matt D Clark, Conceptualization, Supervision, Funding acquisition, Project administration; Neil Hall, Supervision, Investigation, Writing – original draft, Project administration, Writing – review and editing

### Author ORCIDs

Mark McMullan ⓘ https://orcid.org/0000-0002-0711-5666
Lawrence Percival-Alwyn ⓘ https://orcid.org/0000-0001-7725-9203
Rowena Hill ⓘ https://orcid.org/0000-0002-1046-5528
Ross Low ⓘ https://orcid.org/0000-0002-0956-2822
Darren Heavens ⓘ https://orcid.org/0000-0001-5418-7868
Ned Peel ⓘ https://orcid.org/0000-0002-7429-7009
David Swarbreck ⓘ https://orcid.org/0000-0002-5453-1013
Matt D Clark ⓘ https://orcid.org/0000-0002-8049-5423
Neil Hall ⓘ https://orcid.org/0000-0003-2808-0009

### Decision letter and Author response

Decision letter https://doi.org/10.7554/eLife.91245.sa1
Author response https://doi.org/10.7554/eLife.91245.sa2

## Additional files

### Supplementary files

Supplementary file 1. Tables referenced in genome annotation methods. *Genome* – Assembly quality metrics such as N50. *Reads Alignment* – Mapping quality for the 12 RNA-Seq PE read libraries that were aligned to the genome. *Transcript Assemblies* – Transcript assembly metrics for two assembly methods. *Repeats* – Repeat masked regions for protein alignments and the gene build. *Protein Alignments* – Protein alignment summary from 26 representatives (including references). *Mikado Transcript* – Mikado transcript assembly integration gene model statistics. *Augustus Training* – Augustus training results based on Mikado Gold transcripts. *Augustus* – Gene model statistics from three Augustus runs using different levels of evidence. *Minos Release* – Final gene model statistics after selection using Minos-Mikado

Supplementary file 2. Tables detailing sample locations and proximity metrics. *Sample* – Isolate sampling data listed north to south including information on whether isolation was from a wild or crop beet (W-Ag_host) and how isolates clustered after DAPC analysis of genotypes (W-Ag_path-DAPC). Region abbreviation and colour (W-Ag_host) is used in samples names in *Figure 1* in the main text. *Proximity* – Individual host proximity measures are listed for each sample in addition to average and greatest distances per site. Wild sites are sampling locations in the truest sense, but crop samples were grouped by region after being sent from growers via the BBRO. GPS coordinates of sites were not able to be shared due to data protection.

Supplementary file 3. Table of skim sequencing read count and coverage. Skim sequencing of 404 rust peels was used to assess library performance. Rust libraries in green were considered for re-sequencing.

Source data 1. Table of beet rust SNP diversity, differentiation, and annotation data. Gene diversity and differentiation analyses combined with annotation statistics for all genes. Analyses are replicated using SNP diversity from isolates partitioned into five groupings (see 'Pop' column). The groups contain diversity from all individuals combined (allind), crop isolates, defined both using population differentiation analyses (agri-dapc) and based on host affiliation (agri-host) and the wild rusts by both of those criteria (wild-dapc and wild-host). Genetic differentiation measures are recorded between populations defined using the same criteria (-dapc or -host).

MDAR checklist

### Data availability

Illumina paired-end read data for the *U. beticola* genome and 46 re-sequenced isolates as well as 12 RNA-seq libraries used for annotation have been submitted to the European Nucleotide Archive (BioProject PRJNA1194000). The genome, annotation, and variant files are available at Zenodo (https://doi.org/10.5281/zenodo.14288905).

The following datasets were generated:

| Author(s) | Year | Dataset title | Dataset URL | Database and Identifier |
| --- | --- | --- | --- | --- |
| McMullan M, Kaithakottil G, Hill R, Sawford K, Warring S, Kroboth J, Swarbreck D, Percival Alwyn L, Gardiner LJ, Yvanne H, Ward S, Heavens D, Stevens M, Clark M | 2024 | Developing a crop- wild- reservoir pathogen system to understand pathogen emergence and evolution | https://doi.org/10.5281/zenodo.14288905 | Zenodo, 10.5281/zenodo.14288905 |
| McMullan M, Kaithakottil G, Hill R, Sawford K, Warring S, Kroboth J, Swarbreck D, Percivial Alwyn L, Gardiner LJ, Yvanne H, Ward S, Heavens D, Stevens M, Clark M, Hall N | 2024 | Beet rust population genomics and RNA-seq for annotation | https://www.ncbi.nlm.nih.gov/bioproject/PRJNA1194000 | NCBI BioProject, PRJNA1194000 |

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

# Appendix 1

**Appendix 1—table 1.** Sample mapping depth & coverage to *U. beticola* and *B. vulgaris* genomes.

| Sample | Rust mean depth | Host mean depth | Host coverage (%) |
| --- | --- | --- | --- |
| ub-a_037_ofd | 21.8202 | 2.14711 | 46.7026 |
| ub-a_053_hfs | 15.368 | 1.72821 | 24.9724 |
| ub-a_068_hfs* | 9.03836 | 1.90179 | 31.105 |
| ub-a_096_hfs | 26.2938 | 1.61075 | 1.3485 |
| ub-a_207_hsw | 20.6778 | 2.06536 | 44.6831 |
| ub-a_236_bos | 10.1002 | 1.77634 | 8.49367 |
| ub-a_245_gbm | 38.0887 | 1.77758 | 28.4823 |
| ub-a_263_wxm | 19.9234 | 5.58378 | 80.5106 |
| ub-a_278_bos | 25.3881 | 1.82786 | 11.7332 |
| ub-a_290_kln | 19.5187 | 1.78355 | 29.8596 |
| ub-a_292_gbm | 20.9726 | 2.33322 | 54.2479 |
| ub-a_313_gbm | 19.8858 | 1.50272 | 23.9464 |
| ub-a_331_wxm | 15.8241 | 1.54566 | 16.5265 |
| ub-a_338_wxm | 21.0897 | 1.33746 | 9.27185 |
| ub-a_360_bos | 16.2768 | 1.98573 | 4.37071 |
| ub-a_372_kln | 21.0559 | 2.28563 | 52.6314 |
| ub-a_382_lcn | 14.6549 | 1.50788 | 11.3138 |
| ub-a_383_lcn | 12.6089 | 1.53811 | 5.93037 |
| ub-a_400_lcn | 14.6035 | 1.3147 | 6.13931 |
| ub-a_411_bnm | 18.9184 | 1.5897 | 3.77169 |
| ub-a_491_ofd | 15.7196 | 1.63696 | 2.60739 |
| ub-a_509_kln | 18.721 | 1.44109 | 15.5718 |
| ub-a_518_hsw | 28.3377 | 2.8507 | 62.6088 |
| ub-a_519_ofd | 16.328 | 1.45538 | 18.2932 |
| ub-a_527_hsw | 19.5824 | 1.74869 | 30.5799 |
| ub-w_052_swd | 23.8665 | 3.84718 | 67.7464 |
| ub-w_075_ofd | 11.0258 | 1.90397 | 24.3458 |
| ub-w_077_ofd | 10.8219 | 1.6338 | 12.9038 |
| ub-w_090_ofd | 27.6185 | 1.98827 | 30.6426 |
| ub-w_097_swd | 20.835 | 2.04838 | 33.8948 |
| ub-w_109_swd | 17.4465 | 1.64594 | 19.2983 |
| ub-w_142_kln | 12.8499 | 1.74908 | 15.4017 |
| ub-w_145_ofd | 10.4921 | 1.72311 | 14.4469 |
| ub-w_170_hfs | 20.2263 | 1.60315 | 19.9984 |
| ub-w_174_hfs | 15.1471 | 2.0296 | 41.0021 |
| ub-w_175_hfs* | 9.36029 | 1.84945 | 33.1799 |
| ub-w_204_ofd | 15.2831 | 1.83267 | 27.6687 |

*Appendix 1—table 1 Continued on next page*

*Appendix 1—table 1 Continued*

| Sample | Rust mean depth | Host mean depth | Host coverage (%) |
|---|---|---|---|
| ub-w_214_ofd* | 5.50968 | 2.11387 | 36.7496 |
| ub-w_228_swd | 20.314 | 1.55218 | 16.2145 |
| ub-w_229_swd | 23.3174 | 1.57348 | 20.5359 |
| ub-w_418_kln | 14.6769 | 1.72284 | 28.2558 |
| ub-w_419_swd | 17.6829 | 1.60648 | 25.5855 |
| ub-w_451_kln | 21.8048 | 1.39927 | 5.72318 |
| ub-w_458_bos | 23.4803 | 3.91153 | 69.5666 |
| ub-w_467_bos* | 4.75461 | 1.49557 | 12.129 |
| ub-w_470_bos | 14.9618 | 1.56391 | 9.34053 |

*Rust isolates with less than 10 x mean depth were removed from further analyses

# Appendix 2

## Admixture analysis

### Introduction

Network, PCA, DAPC, and admixture methods (main text) suggest first that there are two rust populations and second that five northern isolates sampled on wild hosts, cluster throughout the crop infecting population and apart from all other wild infecting isolates. Here, we look at the impact of increasing $k$ values on the relationships of the isolates we sampled. Specifically, we confirm that increasing $k$ clusters, does not partition the five northern wild sampled isolates, into a different clade. We must conclude, therefore, that they are part of the clade that includes all crop infecting isolates.

Here, we broaden admixture analyses (main text) to explore the impact of increasing $k$ values with reference to isolates placement in the network. Furthermore, we apply a machine learning (ML) methodology in attempt to delineate diversity in the five wild infecting but agriculturally assigned isolates. In ML we used the five isolates in the training data inform the ML, and next we removed them. We set a Wild or Agricultural host, binary classification. In the informed training data, we were able to classify four out of five of these as wild. However, after removing these five wild individuals from the training data and using them as unseen test datapoints, we found that they were now classified as agricultural as in former analyses.

Combined these analyses suggest we have two populations, and that the agricultural population, to which all crop infecting isolates belong, includes the five northern wild infecting isolates. This first population is in addition to an exclusively wild population which is differentiated from that crop infecting population. While our agricultural population is not strict, wild reservoir diversity can be used to highlight diversity from the population containing all crop infecting isolates.

### Methods

We used Scikit Learn (version 3.7) for the ML binary classification analysis to predict if a given sample was from a Wild (class 0) or Agricultural host (class 1) (*Auton and McVean, 2007*). The following classifiers were tested: Logistic Regression, Gaussian process, Random Forest, XGBoost, LightGBM, Support Vector Machine (linear kernel), Decision Tree, and K nearest neighbours (KNN). The features used to train the models were the 1,865,259 SNPs for each sample where SNPs were converted into numerical format with Reference Homozygous SNPs (0/0) denoted as 0, Heterozygous SNPs (0/1) as 1, Homozygous SNPs (1/1) denoted as 2 and missing alleles for a SNP (./.) as NA. Next, we removed any SNPs with NA's (813,649 SNPs remaining). The MinMaxScaler was tested to scale the features from 0 to 1 and initially all samples were included in the analysis where at random 80% of the data was selected to be used for training and the remaining 20% was held out for testing. In the second part of our analysis the five wild individuals were removed during model training and testing and later used as unseen test datapoints. For both cases, next, fivefold cross validation was also performed on the training data. We used K-folds for cross validation (n_splits = 5, stratified k-fold). The methods' hyperparameters were optimized using a grid search to test a range of parameters as per our previous work (*Gardiner et al., 2021*). Class weighting was used to account for uneven balance between the classes. We selected the best performing ML models (using best parameters after fine tuning), according to the highest mean F1-score on test set and cross validation.

We next ran a well-known model explanation algorithm called SHAP (*Lundberg and Lee, 2017*) on the best ML model that was generated using the 813,649 SNPs. We used SHAP to show the local explanation for a specific set of samples to reveal the reasoning behind its prediction for each sample i.e., which SNPs most contributed to it being classified Wild or Agri by the model.

### Results

The number of population clusters, as indicated by the ancestral populations entropy criterion (*Appendix 2—figure 1*), was less clear than using the DAPC method (*Figure 2C*, main text) and, to specifically investigate the admixture in our five northern wild infecting agriculturally assigned isolates, we plot individual admixture for $K$=2:4 (*Appendix 2—figure 2*). With increasing $K$ we get increasing nuance in individual assignment. However, the five wild sample individuals cluster within the crop infecting population (*Appendix 2—figure 2A* red border), were not any more admixed with the wild population than any other agricultural assigned isolate. Moreover, with increasing $K$, admixture revealed relatedness patterns that are observed in the network (*Appendix 2—figure 2B*).

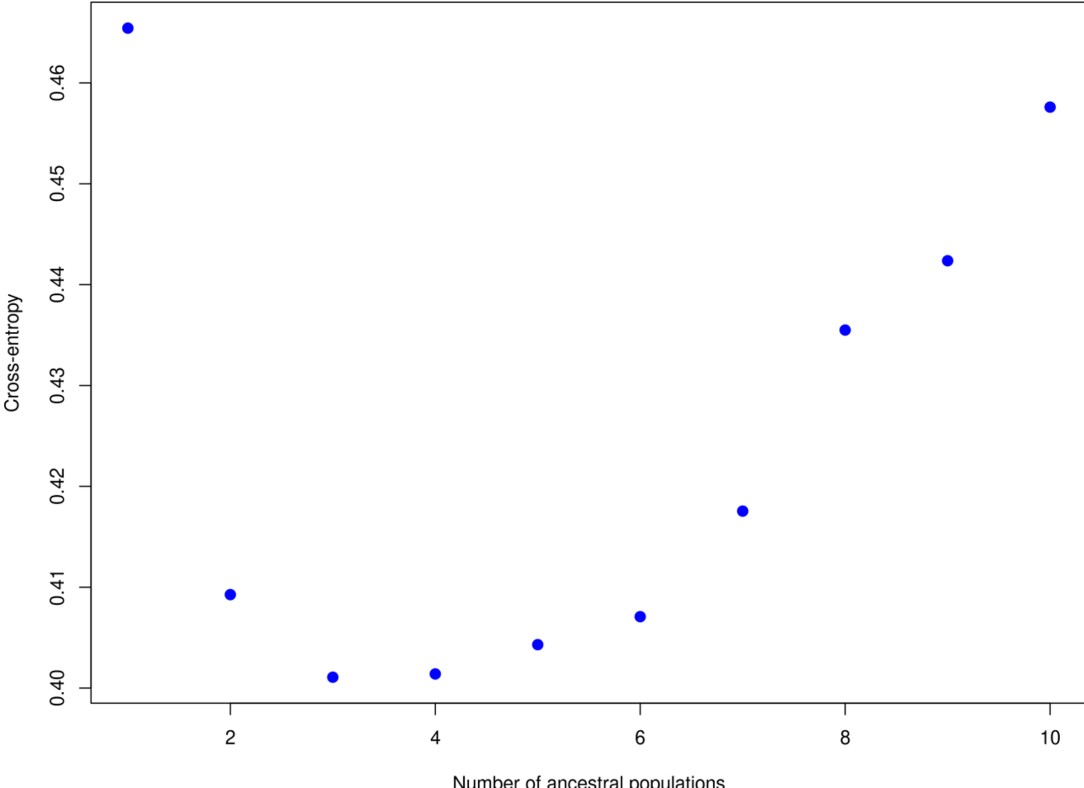

**Appendix 2—figure 1.** The 'knee' in the entropy criterion. Used to define the number of clusters, which here is at two or three. A similar plot produced using DAPC is available in *Figure 2* of the main text which indicates two populations.

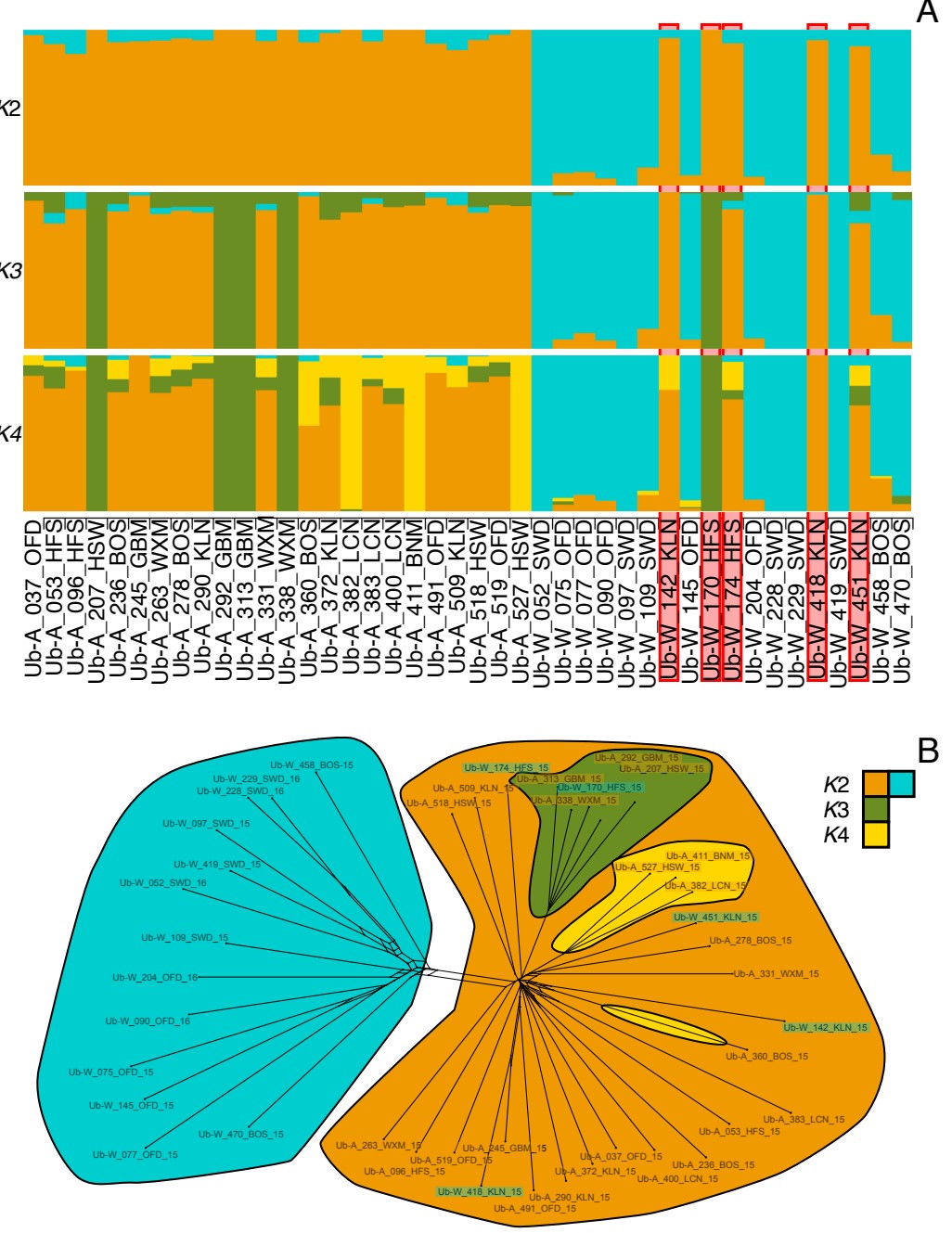

**Appendix 2—figure 2.** Bar plots of individual admixture of rust isolates. (**A**) Admixture proportions for each individual for *K*=2:4. Isolates are ordered in two blocks as being sampled from a crop (Ub-A…) or a wild host (Ub-W…). The five wild sampled individuals that were identified to be within the crop infecting populations are bordered in red. (**B**) Network of relatedness (*Figure 3B* main text) among isolates. Sample name highlighting reflects host affiliation (crop = orange, wild = turquoise). Clade colouring reflects individual majority assignment colour from A.

When we include all samples in the training/testing process, the ML KNN classifier shows the best performance on the test dataset (*Appendix 2—figure 3*). Across the test and training data KNN classified one of the five isolates as Agri that had been sampled on a wild host (*Appendix 2—figure 3b*, sample Ub-W_142_KLN; see *Appendix 2—table 1* for model KNN model details).

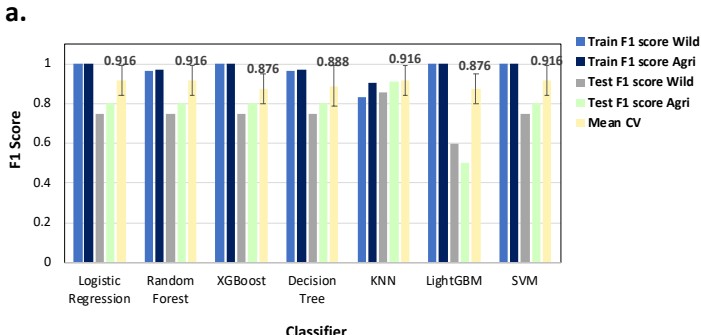

**a.**

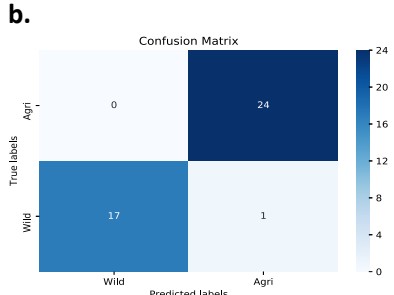

**b.**

**c.**

| Classifier | Train F1 score Wild | Train F1 score Agri | Test F1 score Wild | Test F1 score Agri | Mean CV | SD CV |
|---|---|---|---|---|---|---|
| Logistic Regression | 1 | 1 | 0.75 | 0.8 | 0.916 | 0.076 |
| Random Forest | 0.963 | 0.974 | 0.75 | 0.8 | 0.916 | 0.076 |
| XGBoost | 1 | 1 | 0.75 | 0.8 | 0.876 | 0.074 |
| Decision Tree | 0.963 | 0.974 | 0.75 | 0.8 | 0.888 | 0.102 |
| KNN | 0.833 | 0.905 | 0.857 | 0.909 | 0.916 | 0.076 |
| LightGBM | 1 | 1 | 0.6 | 0.5 | 0.876 | 0.074 |
| SVM | 1 | 1 | 0.75 | 0.8 | 0.916 | 0.076 |

**Appendix 2—figure 3.** Results from ML classification analysis using 813,649 SNPs. For the ML analysis to predict Wild/Agricultural in binary classification: (**a**) Bar charts showing the F1 scores for the training and test datasets (using best parameters) and the mean F1 scores (Mean CV) after fivefold cross validation with the standard deviation shown as error bars. Labels for different bar colours are detailed in legend to the right of the plot. (**b**) Confusion matrix for the KNN ML model for the training + test dataset i.e., all samples. (**c**) Table of performance values for each ML model as shown in (**a**).

We next used the model explanation algorithm SHAP on this KNN model to assess the reasoning behind its predictions for the five wild sampled individuals that were identified to be within the crop infecting populations by the DAPC methods, i.e., which SNPs most contributed to each sample being called Wild or Agri. *Appendix 2—figure 4* shows the results of this investigation where it is clear that the sample Ub-W_142_kln (*Appendix 2—figure 4a*), classified as Agri by the KNN ML model, mainly has SNP alleles that match the profile that the model has learned and associated with an Agri sample, while the four remaining samples (*Appendix 2—figure 4b-e*) Ub-w_170_hfs, Ub-w_174_hfs, Ub-w_418_kln and Ub-w_451_kln that were classified as Wild by the KNN ML model have a mixture of Agri and Wild SNP alleles in the top 10 most impactful that the ML model used for classification. We then repeated this analysis, this time removing the five northern wild infecting isolates from the ML model during training, testing and tuning. The resultant best performing trained and tuned ML model was also a KNN model that showed perfect F1 scores (of 1.000) across the training and test data and during cross validation. However, when the five northern wild infecting isolates were used as an unseen test set, this model labelled all five of the samples as Agri as per the DAPC methods. Considering this, these wild samples have the hallmarks of 'incorrectly labeled' samples (or agri samples) when being considered in a binary classification. We used the model explanation algorithm SHAP on this KNN model to assess the reasoning behind its predictions for the five wild sampled individuals.

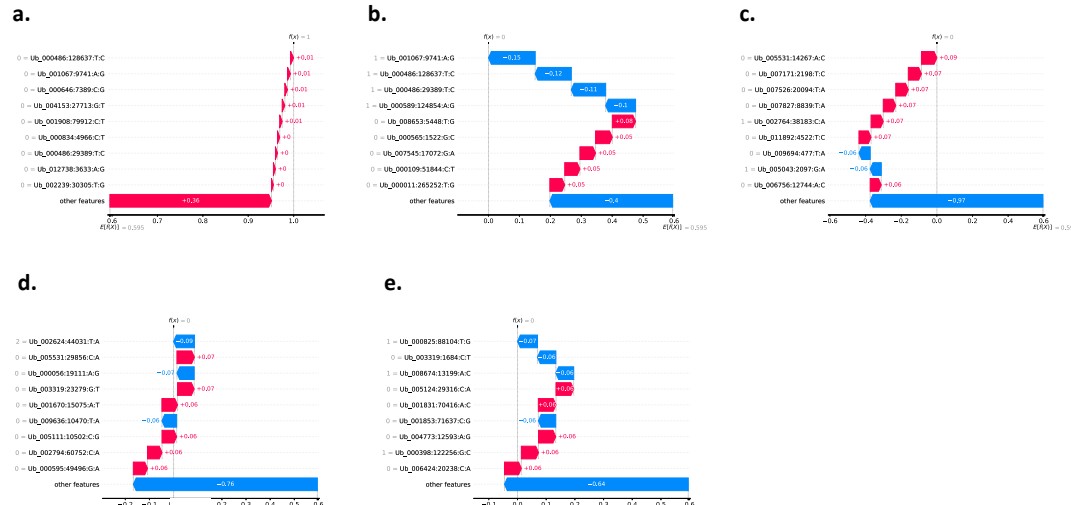

**Appendix 2—figure 4.** Using SHAP to explain our KNN binary classification model using 813,649 SNPs. Here, we show the local explanation for the prediction of the five wild sampled individuals (**a**) Ub-w_142_kln that the model predicted as class 1=Agri. (**b**) Ub-w_170_hfs that the model predicted as class 0=Wild. (**c**) Ub-w_174_hfs that the model predicted as class 0=Wild. (**d**) Ub-w_418_kln that the model predicted as class 0=Wild. (**e**) Ub-w_451_kln that the model predicted as class 0=Wild. All plots (**a–e**) show the top 10 most impactful SNP alleles and their respective impact scores (most impactful ordered from top to bottom) that contributed to the classification of each sample as class 1 (Agri) as calculated using SHAP. The red colours denote an impact score for a SNP allele that has positively impacted the classification of the sample i.e., if we sum all SNP allele impact scores for a sample, a more positive score tending towards 1 gives the sample a classification of class 1 (Agri). The blue colours denote an impact score for a SNP allele that has negatively impacted the classification of the sample i.e., if we sum all SNP allele impact scores for a sample, a more negative score tending towards 0 gives the sample a classification of class 0 (Wild). Sample (**a**) Ub-w_142_kln was called class 1 (Agri) by the model and samples (**b–e**) Ub-w_170_hfs, Ub-w_174_hfs, Ub-w_418_kln, Ub-w_451_kln were called class 0 (Wild) by the model. In grey next to each SNP denotes the specific allele that the sample has i.e., Reference Homozygous SNPs (0/0) denoted as 0, Heterozygous SNPs (0/1) as 1, Homozygous SNPs (1/1) denoted as 2.

**Appendix 2—table 1.** Best performing classification models from ML analyses.

Detailing the parameter sets used for our best performing models.

| Purpose | Feature set | classifier | HYPERPARAMETERS |
|---|---|---|---|
| Binary classification Wild/Agri | 813,649 SNPs | KNN | Pipeline (memory = None, steps=[('clf', KNeighbors Classifier (algorithm='auto', leaf_size = 1, metric='minkowski', metric_params = None, n_jobs = 2, n_neighbors = 18, p=2, weights='distance'))], verbose = False) |

# Appendix 3

## Gene coverage

### Introduction

Gene presence and absence is an important phenomenon that could be subject to selection in different populations. Gene presence absence is better estimated using a highly contiguous pangenomics approach and here we use short-read genome re-sequencing and mapping to a single reference. However, to identify regions of the genome present in all isolates, we estimated gene presence and absence using 60% gene coverage as a cut-off. Here, we also demonstrate the difference in the length distribution of the CDS regions of non-effector and effector genes.

### Results

We found that 87 genes were missing in one or more isolates but that 48 of those were missing in all isolates (*Appendix 3—table 1*). Seven genes were missing in one but not the other population, however, none of those were fixed in the population they were observed in. UROBE1963355_EIv1.0_0143750, …0143760 & …0143770 are adjacent and found in nine agricultural isolates and have unknown and functions (InterPro and gene ontology, see Supplementary Information 9 for gene function). UROBE1963355_EIv1.0_0151510 & …0151520 are found in just three and four (respectively) agricultural isolates and have functions associated with nuclease activity. UROBE1963355_EIv1.0_0155690 is present in just three agricultural isolates and has InterPro and Gene Ontology functions associated with membrane transport and UROBE1963355_EIv1.0_0155880, also present in three isolates has a PKD/Chitinase domain, often involved in interactions with other proteins. While present in the agricultural population but not the wild population, these seven genes were all at low to medium frequencies, which is consistent with the expectation based on the frequency of gene presence based other genes present in both populations (*Appendix 3—figure 1*).

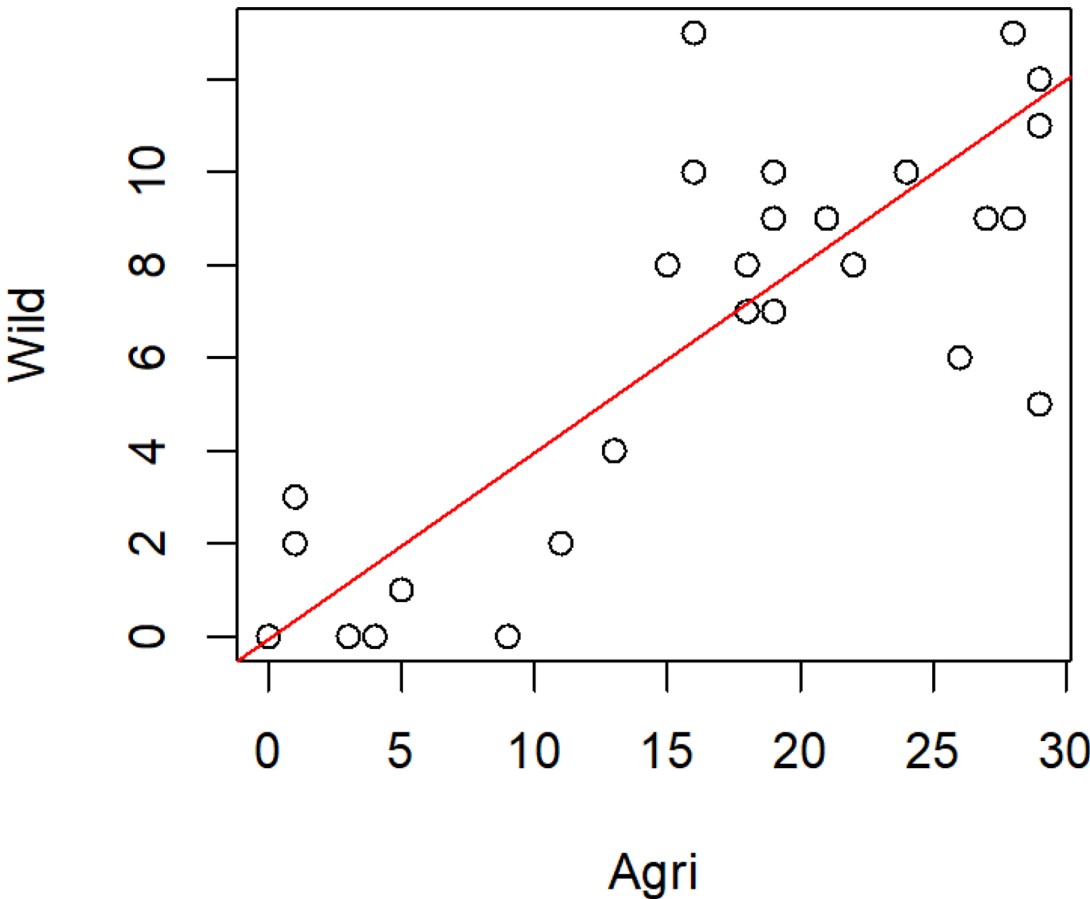

**Appendix 3—figure 1.** Missingness of genes from both the wild and agricultural populations (populations defined using DAPC). For all non-effector genes that are missing in any one isolate, we present the number of isolates containing that gene in each population ($r^2$=0.89 p=<2.2e-16). Most genes (48) are absent both populations.

The signature of missingness in effectors is also dominated by effectors that were present in the assembly but absent from all isolates (four effectors: UROBE1963355_EIv1.0_0119130, …0119140, …0146660 & …0146690). Just a single additional effector was present in some but not all isolates (UROBE1963355_EIv1.0_0024320). This effector was absent from one agricultural isolate and four wild isolates (ub-a491_ofd_15, ub-w_090_ofd_16, ub-w_097_swd_15, ub-w_109_swd_15 & ub-w_145_ofd_15).

Here, we include the lengths of gene CDS regions. Effectors are shorter on average than non-effector genes (*Appendix 3—figure 2*; median CDS length: non-effector=951, effector = 534; Wilcoxon W=1189236, p<0.001).

**Appendix 3—table 1.** Numbers of genes with individual coverage >=60%.

| Isolate | All genes (including TEs) | % | Genes | % |
|---|---|---|---|---|
| ub-a_037_ofd | 15536 | 99.6 | 9084 | 99.3 |
| ub-a_053_hfs | 15539 | 99.7 | 9088 | 99.3 |
| ub-a_096_hfs | 15535 | 99.6 | 9087 | 99.3 |
| ub-a_207_hsw | 15536 | 99.6 | 9084 | 99.3 |
| ub-a_236_bos | 15532 | 99.6 | 9081 | 99.3 |
| ub-a_245_gbm | 15537 | 99.6 | 9086 | 99.3 |
| ub-a_263_wxm | 15534 | 99.6 | 9086 | 99.3 |

*Appendix 3—table 1 Continued on next page*

*Appendix 3—table 1 Continued*

| Isolate | All genes (including TEs) | % | Genes | % |
|---|---|---|---|---|
| ub-a_278_bos | 15537 | 99.6 | 9087 | 99.3 |
| ub-a_290_kln | 15538 | 99.6 | 9086 | 99.3 |
| ub-a_292_gbm | 15537 | 99.6 | 9085 | 99.3 |
| ub-a_313_gbm | 15539 | 99.7 | 9087 | 99.3 |
| ub-a_331_wxm | 15540 | 99.7 | 9090 | 99.4 |
| ub-a_338_wxm | 15542 | 99.7 | 9090 | 99.4 |
| ub-a_360_bos | 15531 | 99.6 | 9082 | 99.3 |
| ub-a_372_kln | 15535 | 99.6 | 9083 | 99.3 |
| ub-a_382_lcn | 15540 | 99.7 | 9089 | 99.4 |
| ub-a_383_lcn | 15538 | 99.6 | 9086 | 99.3 |
| ub-a_400_lcn | 15538 | 99.6 | 9086 | 99.3 |
| ub-a_411_bnm | 15539 | 99.7 | 9088 | 99.3 |
| ub-a_491_ofd | 15531 | 99.6 | 9083 | 99.3 |
| ub-a_509_kln | 15536 | 99.6 | 9086 | 99.3 |
| ub-a_518_hsw | 15536 | 99.6 | 9084 | 99.3 |
| ub-a_519_ofd | 15543 | 99.7 | 9091 | 99.4 |
| ub-a_527_hsw | 15541 | 99.7 | 9090 | 99.4 |
| ub-w_052_swd | 15522 | 99.5 | 9083 | 99.3 |
| ub-w_075_ofd | 15526 | 99.6 | 9078 | 99.2 |
| ub-w_077_ofd | 15523 | 99.6 | 9082 | 99.3 |
| ub-w_090_ofd | 15528 | 99.6 | 9083 | 99.3 |
| ub-w_097_swd | 15529 | 99.6 | 9083 | 99.3 |
| ub-w_109_swd | 15527 | 99.6 | 9081 | 99.3 |
| ub-w_142_kln | 15538 | 99.6 | 9088 | 99.3 |
| ub-w_145_ofd | 15533 | 99.6 | 9082 | 99.3 |
| ub-w_170_hfs | 15533 | 99.6 | 9081 | 99.3 |
| ub-w_174_hfs | 15537 | 99.6 | 9085 | 99.3 |
| ub-w_204_ofd | 15525 | 99.6 | 9078 | 99.2 |
| ub-w_228_swd | 15536 | 99.6 | 9090 | 99.4 |
| ub-w_229_swd | 15533 | 99.6 | 9087 | 99.3 |
| ub-w_418_kln | 15535 | 99.6 | 9084 | 99.3 |
| ub-w_419_swd | 15533 | 99.6 | 9085 | 99.3 |
| ub-w_451_kln | 15533 | 99.6 | 9084 | 99.3 |
| ub-w_458_bos | 15524 | 99.6 | 9083 | 99.3 |
| ub-w_470_bos | 15529 | 99.6 | 9082 | 99.3 |

Colours reflect origin orange=crop sampled, blue=wild, purple=wild sampled cluster within agricultural.

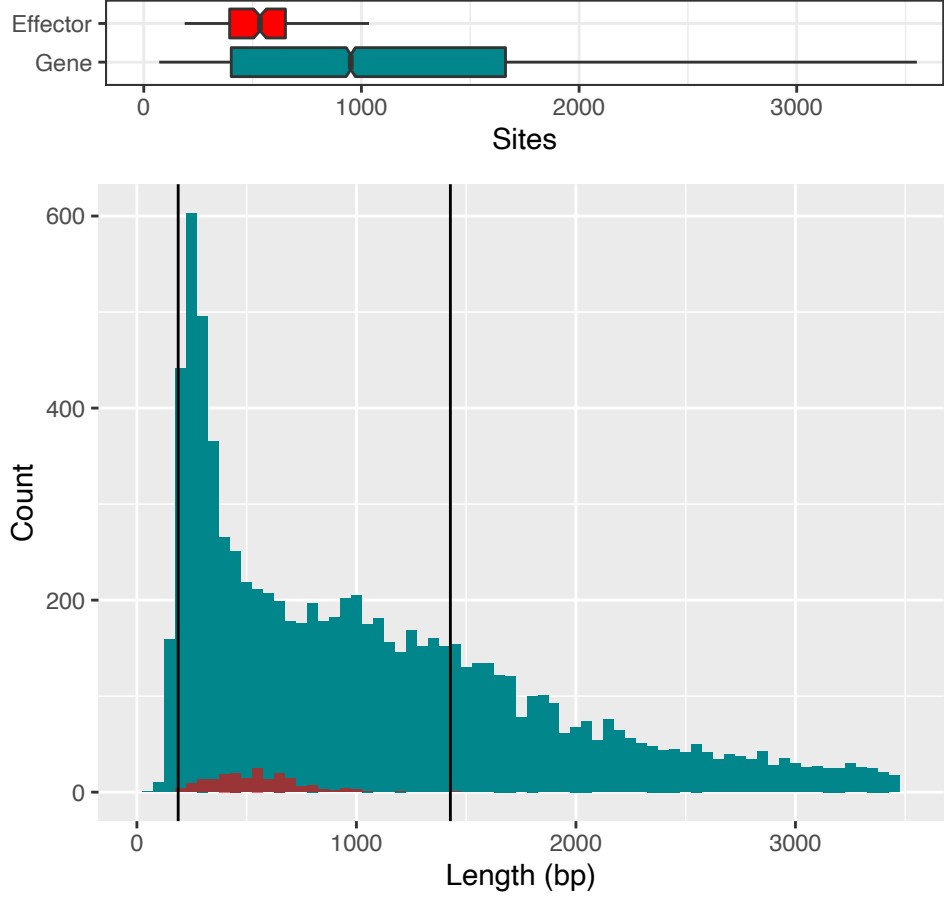

**Appendix 3—figure 2.** Histogram of counts of observed effector and non-effector gene coding sequence (CDS) lengths (between 1-3500 bp) shows counts per gene type. Vertical black lines indicate the effector length range.

# Appendix 4

## Impact of polymorphism, function, and divergence

### Introduction

Those effectors evolving under diversifying selection might be expected to have higher levels of diversity and differentiation than other genes on average. In the main text we present work based on SNPs as measured by nucleotide diversity ($\mu$) and genetic differentiation ($F_{ST}$) which are appropriate for population genetic analyses based on more fragmented genomes. However, these methods don't specifically account for haplotype diversity. Here, we build on the methods used in the main text, first to better understand the impact of expression mediation of virulence, then the impact genetic divergence ($d_{XY}$) on effectors. We then use methods impacted to a greater degree by linkage to look for signals of selection (selective sweeps) and recombination. Not only are these linkage-based methods hindered by a fragmented repeat rich genome but additionally by invasion demography which can produce signals similar to a selective sweep (*Pavlidis and Alachiotis, 2017*). We do not find a signal consistent with mediation of virulence in effectors by expression. However, in analyses of divergence and selective sweeps we find some evidence of increases in both signals associated with effectors. Our assessment of recombination shows a somewhat contradictory pattern in this repeat rich dikaryon. Specifically, that the agricultural population has a higher rate of recombination than the wild.

### Results and discussion

Effector silencing and alternative splicing could be an important, rapidly evolving mechanism of pathogen adaptation (*Jeon et al., 2022*). Here, using genome polymorphism data, we observe indeed that a greater proportion of non-effector genes are alternatively spliced ($\chi2$: 198, df=1, p=0.03; *Appendix 4—figure 1*). The partitioning of SNP and indel variants in genes and effectors did not appear to differ among gene types or populations. High impact mutations responsible for start lost, stop gained or stop lost impacted a very low proportion of gene types again with no consistent pattern.

Effectors retain a signal of diversifying selection, when compared between wild and crop populations (see main text). This signal is consistent with the idea that selection for alternative variants in these two populations, is present at enough effector genes to significantly increase the differentiation measure ($F_{ST}$) on average. However, this measure is based on allele frequency distributions, absolute sequence divergence ($d_{XY}$) is an independent measure used to indicate divergence, for a given region of the genome. To determine whether effector divergence is greater than that of non-effector genes, we re-called all positions in the genome to reduce bias caused by missing sites *Korunes and Samuk, 2021* and measured divergence and differentiation.

Both differentiation and divergence were estimated to be significantly greater at effectors than non-effector genes (median $F_{ST}$: Effector=0.103, non-effector Gene=0.085; Wilcoxon($\pi$) W=751402, p=0.001; median $D_{XY}$: Effector=0.3×10$^{-3}$, non-effector Gene=0.2×10$^{-3}$; Wilcoxon($\pi$) W=818110, p=0.031; *Appendix 4—figure 2*). For genetic differentiation this recapitulates the result in the main text, although here using the all-sites methodology. The measure of absolute divergence is also significantly greater for 10kb windows that span an effector gene.

Divergence ($D_{XY}$), is often used as evidence for regions of the genome important for speciation, where divergence accumulates in regions important for ecological specialisation or reproductive isolation (*genomic islands of speciation*) (*Cruickshank and Hahn, 2014*). Genomic regions resistant to gene flow would be expected to have higher absolute and relative measures of species divergence, as we observe in the present study. At the genome level however, we do not expect a trajectory of divergence leading towards speciation, and so divergent haploblocks are unlikely to extend from ecologically specialised haplotypes. To assess the broader patterns of divergence at the genome scale we plotted windows of divergence and differentiation.

Perhaps because of the fragmented nature of the genome assembly, we do not observe broader fluctuations in divergence and differentiation across many of these contigs, the majority of which contain fewer than ten independent windows (see *Appendix 4—figure 3*). While the largest contig in the assembly is over 500kb, it contains just six gene annotations, none of which are effectors. The mean number of genes present on a contig (with at least one gene) is just 2.3, and including those contigs greater than 200kb, this increases to 5.0 genes. Analysis of genomes at this scale favour gene level analyses (see main text), as short contigs prevent analysis of clusters of genes (or

effectors), as well as capturing fluctuations in broader signatures around those regions. Moreover, such signals are usually observed between isolates over broader geographic, even global scales (*Gladieux et al., 2015b*), making our observation of both increased relative and absolute divergence at effector genes perhaps more remarkable.

In line with the observation that genetic divergence is greater in effectors, is the observation that the signal of selective sweeps is also greater in effectors, and that this signal is present in agricultural effectors but not wild ones (relative to non-effector genes; see main text), consistent with selection operating on effectors. However, the signal of selective sweeps is not so great in any one effector to take it over the 99% threshold and so, again, these observations must be taken in consideration of the fragmented genome and the underlying demography, in which we expected invasion dynamics to also limit levels of diversity across the genome, particularly in the crop infecting population.

In the main text we use $F_{IS}$ to explore evidence for differing rates of clonality between the wild and agricultural populations and we find that the crop infecting population has significantly lower (and negative) levels of diversity which is consistent with the maintenance of heterozygosity in clonal lineages (see main text). Here, we used LDhat to identify evidence for varying rates of recombination among populations (*Table 1*). Surprisingly we find that the crop infecting population has a higher rate of recombination, on average (and in 9 out of 10 contigs), than that of the wild population (mean $4N_er$: agri=196.5, wild=90.1l *Appendix 4—table 1*). Reconciling these results with $F_{IS}$ results from the main text is difficult without a high contiguity phased assembly. Heterozygosity-based measures (e.g. $F_{IS}$) of modes of reproduction are useful in the present case because they are biased to a lesser degree by the quality and contiguity of a genome assembly. Here, we have a highly fragmented unphased dikaryotic assembly. Our linkage-based statistics suggest the opposite is true, that rates of recombination are higher in the agricultural population. To further reconcile genotypic and haplotypic indications of the modes of reproduction operating in these two populations, we must generate de novo phased assemblies, from each population.

## Methods

Absolute genetic divergence requires an all sites vcf file to properly account for (and distinguish) missing sites from zero polymorphism between haplotypes. Reads were remapped to the genome and the 'all sites' vcf was filtered as in the main text (max depth 69, 1.8x mean overall depth) which resulted in 1,987,847 SNPs. Diversity and divergence analyses in sliding windows (https://github.com/simonhmartin/genomics_general; *Martin, 2025*) was used to calculate both genetic diversity ($F_{ST}$) and divergence ($D_{XY}$) in 10kbp windows (1kbp slide). Bedtools-v2.31.0 annotate (*Quinlan and Hall, 2010*) was used to extract the best overlapping window for gene analyses. Analysis of genetic differentiation was used (along with others, which are not shown) to confirm the result in the main text using the original vcf (*Appendix 4—figure 2*).

For each of the 10 largest scaffolds, VCFtools (--ldhat-geno) was used to generate the sites and locs file formats required by LDhat v2.2 (*Auton and McVean, 2007*). Likelihood lookup tables were created using LDhat complete with parameters set to -rhomax 100 and -n_pts 101. This was performed separately for agricultural and wild clades, using 58 and 26 sequences (-n) and respective theta values (-theta) of 0.000324 and 0.000358. To obtain the population-scaled recombination rayes ($4N_er$), LDhat's pairwise module was used, following the method described by *Wakeley, 1997*.

**Appendix 4—table 1.** Recombination estimates from analysis of the ten largest contigs from crop infecting (agri) and wild rust populations.

| Contig | Agri ($4N_er$) | Wild ($4N_er$) |
|---|---|---|
| 1 | 93.096 | 75.502 |
| 2 | 40.023 | 19.438 |
| 3 | 179.059 | 48.146 |
| 4 | 71.138 | 28.805 |
| 5 | 1104.255 | 235.567 |
| 6 | 91.213 | 62.207 |

*Appendix 4—table 1 Continued on next page*

*Appendix 4—table 1 Continued*

| Contig | Agri (4N$_e$r) | Wild (4N$_e$r) |
|---|---|---|
| 7 | 275.848 | 171.298 |
| 8 | 12.372 | 7.033 |
| 9 | 65.892 | 39.025 |
| 10 | 31.94 | 214.332 |

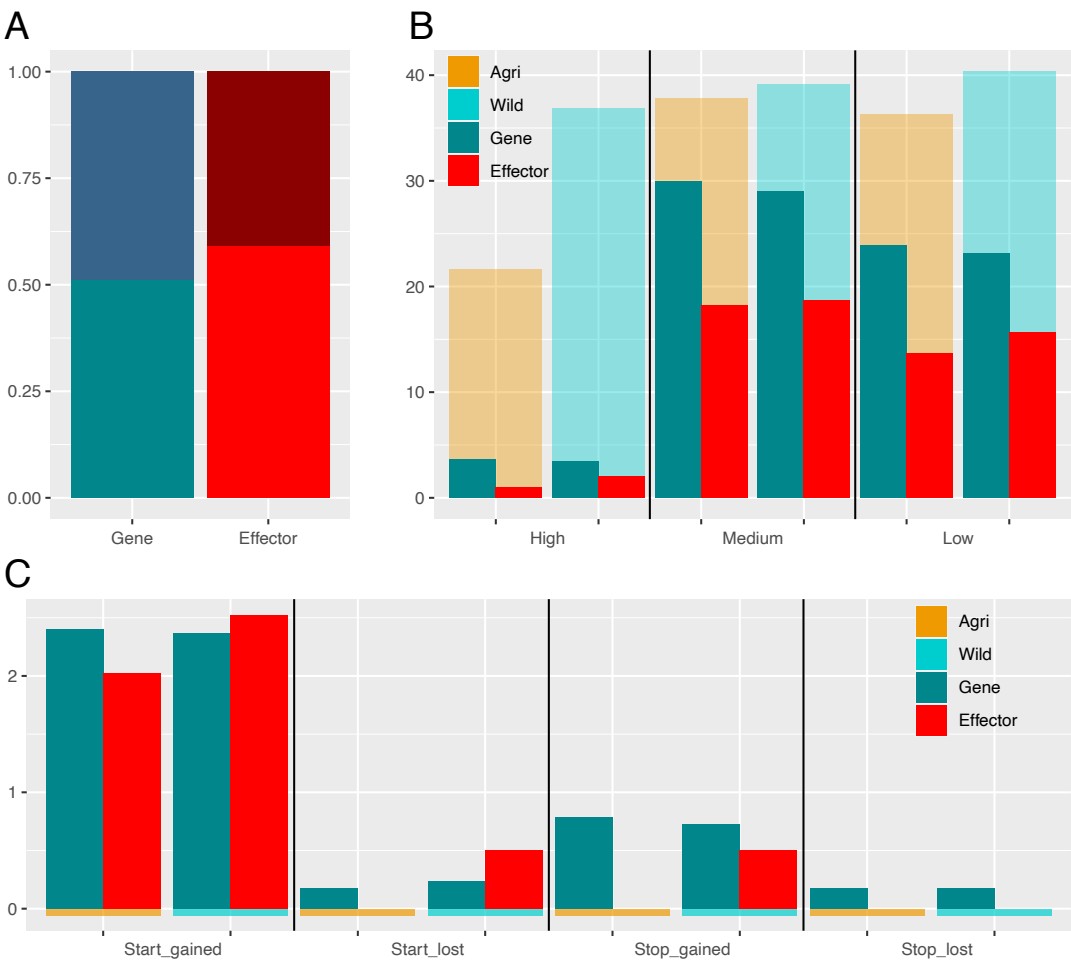

**Appendix 4—figure 1.** Alternative splicing and variant impacts in non-effector genes and effectors. (**A**) Proportions of alternatively spliced genes is greater than that of effectors (shaded region is alternatively spliced). (**B**) Percentages of Genes (turquoise) and Effectors (red) with High, Medium, and Low impact variants (SNPs and indels combined). As expected based on average gene length there are a greater percentage of non-effector genes with variants (in all classes). Orange and blue backed bars reflect the percentage of mutated effectors relative to genes in each population. While a greater percentage of genes carry a polymorphism, the relative proportion of effectors that carry a polymorphism is always higher in the wild, compared to agricultural. (**C**) Percentages of genes and effectors impacted by start and stop, gained and lost polymorphisms.

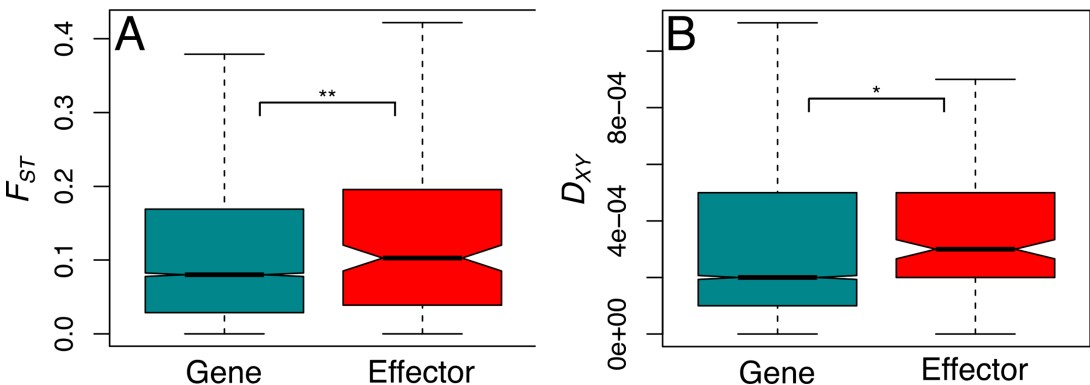

**Appendix 4—figure 2.** Genetic differentiation and divergence are significantly greater at effector loci. Genetic differentiation ($F_{ST}$, **A**) and genetic divergence ($D_{XY}$, **B**) present at effector and non-effector genes, reanalysed using 10 kb windows across all-sites. Genetic diversity and differentiation, are both significantly greater in effectors than non-effectors (see main text) which suggests that more genes in this group are exposed to diversifying selection, favouring effector polymorphism specific to each environment.

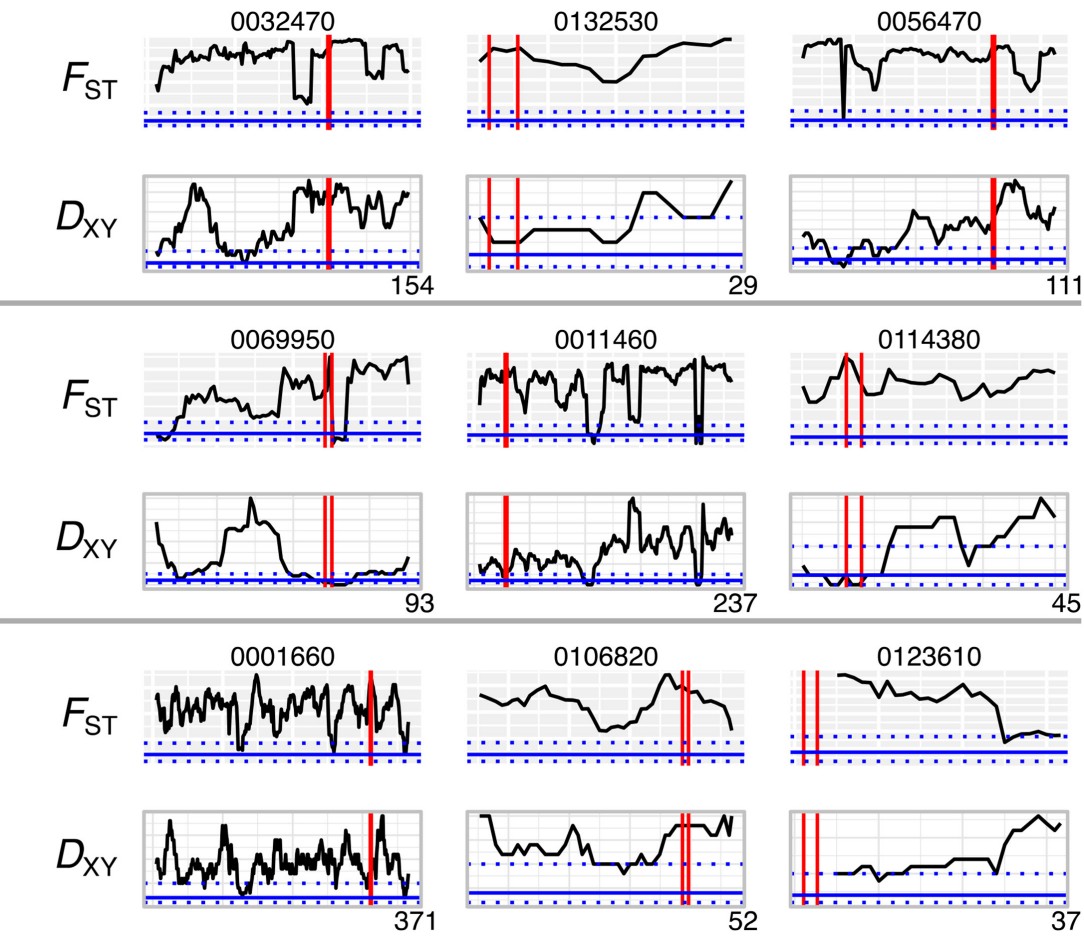

**Appendix 4—figure 3.** Genetic differentiation ($F_{ST}$) and divergence ($D_{XY}$) for most differentiated effectors. Windows of 10 kb across contigs containing effectors with the highest $F_{ST}$ values, shown above (shaded background) and $D_{XY}$ below (clear background). Vertical red lines show the start and end position of the effector, and horizontal blue lines show the median and interquartile range for all genes. Contig lengths shown on x-axis (kb).

## Appendix 5

### Host partitioned analyses

Introduction

Here, we replicate and complement analyses from the main text for groups of isolates as defined by the *host* they were sampled from, as opposed to the *population* they were attributed to using network and PCA (DAPC) methods (see main text). Using DAPC and network methods, five of 18 wild-host sampled isolates were assigned to the population that contained all agricultural isolates and here we analyse those 18 isolates sampled from a wild-host against 24 isolates sampled from an agricultural-host.

Results and discussion

Allele frequency distributions are impacted by processes such as population expansion and contraction as well as varying modes of sexual/clonal reproduction. Distinguishing these processes in the present system is difficult because we believe they may be interacting. For example, clonal reproduction is expected to preserve mutations in a heterozygous state and each polymorphism is expected to be observed at a low frequency (rare). This sets up the expectation that our agricultural population, which we predict has a greater rate of clonal to sexual reproduction, has a greater proportion of rare variants. However, we may also observe boom and bust cycles in this population because of recolonisation after crop free periods. These annual reduction and clonal expansions would preserve those heterozygous mutations and after expansion, they would be shared among all individuals of that linage, having the opposite effect. The pattern we observe in the present study is not clear cut and requires a better understanding of the rates of sexual reproduction (linkage) as well as temporal estimates of genetic differentiation in order to parameterise an individual based model and test our predictions against these observations.

First, we consider DAPC population. In the low allele frequency category, as expected we observe a larger number of rare variants in the agricultural population (*Appendix 5—figure 1*). However, agricultural and wild populations don't deviate from each other in most of their minor allele frequency bins, nor in Tajima's D. The increase in proportion of rare variants (increased clonal reproduction) in agriculture would have the impact of reducing Tajima's D in the bins below zero. The DAPC agricultural population does dominate across this region but so too does it at higher Tajima's D values (expansion from a bottleneck). Moreover, the DAPC wild and agricultural Tajima's D distributions largely overlap.

The overall pattern of minor allele frequency and Tajima's D distributions are somewhat similar when we consider individuals grouped by the host from which they were sampled. In this host differentiated set, the agricultural group again retain an increased rare variant frequency although this is much less apparent in the Tajima's D distribution.

Next, to assess the impact of using host assignment as opposed to DAPC assignment (main text), we repeat analyses from the main text for host-grouping and compare to the DAPC-grouping results. The impact of host grouping on genome wide nucleotide diversity (median $\pi$ 10 Kbp windows) is that the wild-host group further increases relative to wild-DAPC group, while on the agricultural side the decline is only marginal (Host $\pi$: Agri $0.251 \times 10^{-3}$; Wild $0.320 \times 10^{-3}$ DAPC $\pi$: Agri $= 0.252 \times 10^{-3}$; Wild $= 0.269 \times 10^{-3}$; *Figure 2A* and main text). This pattern suggests that transferring some agricultural-DAPC polymorphism over to the wild-host group, increases diversity there without reducing diversity in the agricultural host-group because much of the diversity remains in that group. The difference between wild and agricultural host-groups has increased and they remain significantly differentiated (Wilcoxon($\pi$) W=2362353216, p<0.001). The correlation in nucleotide diversity for host-grouping is strong (r2=0.94 p<0.001; *Appendix 5—figure 2B*) and marginally higher than for the DAPC-grouping (r2=0.92 p<0.001; see main text). Again, this is consistent with the expectation, as five individuals from one DAPC defined group have been moved across to another, increasing the apprant correlation in diversity between wild-host and agricultural-host groups.

Analysis of polymorphic gene CDS nucleotide diversity results in the same conclusions as in DAPC defined groups (main text). The increase in wild-host nucleotide diversity is reflected here and increased the significance of agricultural comparisons to the wild, such that wild-host effector diversity (median $\pi=0.482 \times 10^{-3}$) remains significantly higher than agricultural-host effector diversity (median $\pi=0.085 \times 10^{-3}$; Wilcoxon($\pi$) W=1106, p<0.001; *Appendix 5—figure 2C*) and wild-host

non-effectors (median $\pi$=0.110 × 10$^{-3}$) remain higher than agricultural-host non-effectors (median $\pi$=0.054 × 10$^{-3}$; Wilcoxon($\pi$) W=6500805, p<0.001). Within population wild-host effectors remain significantly higher than non-effectors (Wilcoxon($\pi$) W=70749, p<0.001) and agricultural-host non-effectors remain not significantly less than effectors in their nucleotide diversity (Wilcoxon($\pi$) W=111735, p=0.158).

Finally, we consider the impact of host grouping on genetic differentiation ($F_{ST}$). DAPC-grouped $F_{ST}$ values are more than 40% greater than host-group values (non-effectors=1.49, effectors = 1.42) and, again these results are consistent with expectation (*Appendix 5—figure 2D*). The correlation in genetic differentiation at genes, between grouping methods (Host or DAPC) is strong ($F_{ST}$ Non-effectors: r$^2$=0.95, p<0.001; $F_{ST}$ Effectors: r$^2$=0.97, p<0.001; *Appendix 5—figure 2D*). Despite the 40% lower level of genetic differentiation between host-grouped samples a small number of non-effector genes (272) have a greater differentiation in the host-differentiated set than the DAPC-differentiated. However, without an a priori prediction of the groups of genes we expect to find here such as the one we have for effectors, we can't discriminate from chance within this set without temporal sampling.

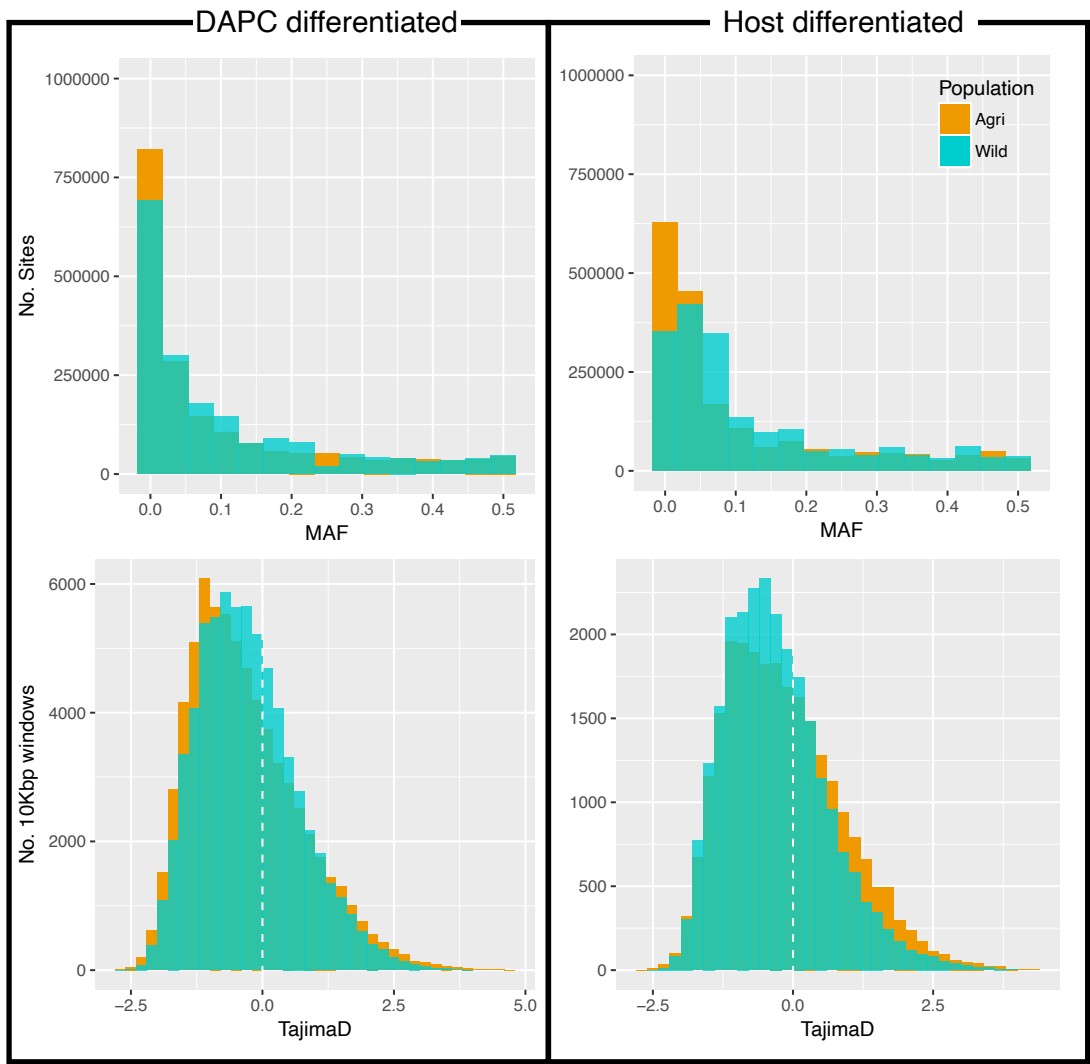

**Appendix 5—figure 1.** Minor allele frequency (MAF, top) and Tajima's D (bottom) for all variants. MAF and Tajima's D analysed twice, once for each of the two methods of grouping these individuals (DAPC, left and Host differentiated, right; see *Source data 1*).

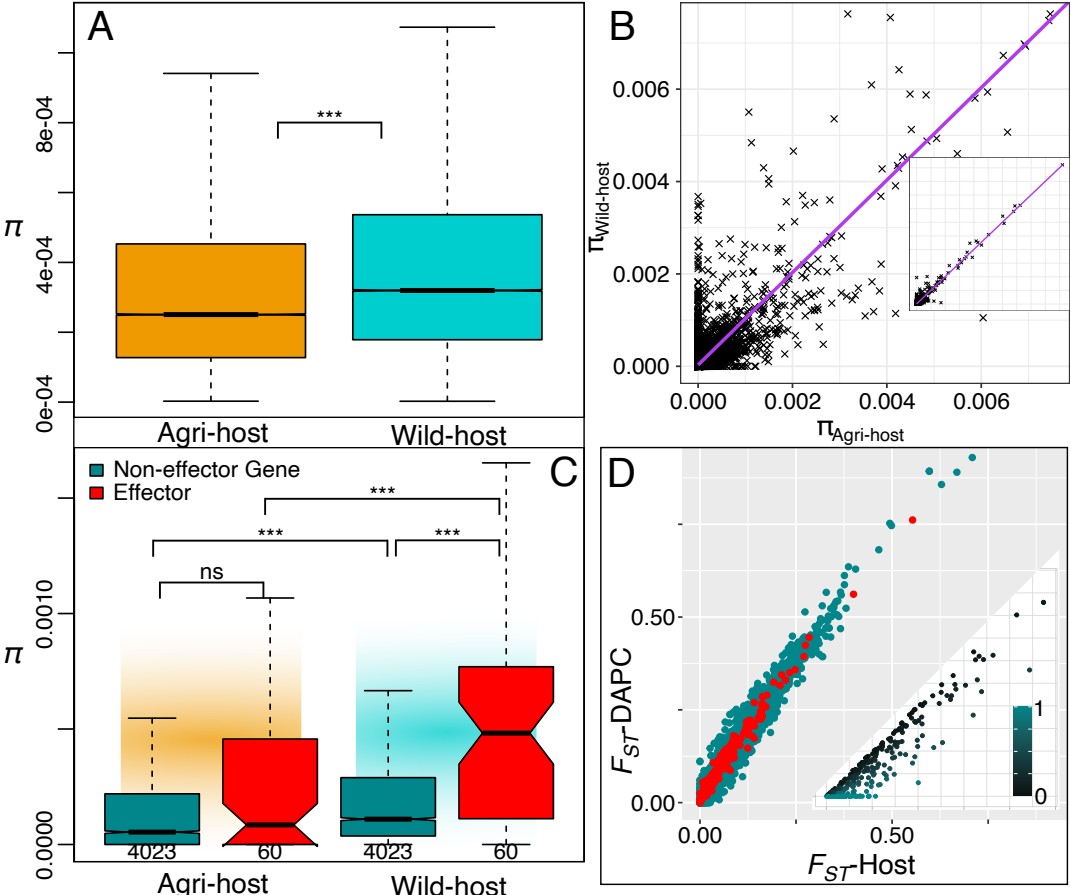

**Appendix 5—figure 2.** Host defined groups retain a signal of adaptive diversity in effectors. (**A**) Genome wide nucleotide diversity (π, 10 kbp non-overlapping blocks) is significantly higher in the wild host-group compared to the agricultural-host group and (**B**) at the gene level, nucleotide diversity within the gene coding sequence (CDS) is strongly correlated between agricultural-host and wild-host groups. The inset shows the full range (0.00-0.05). (**C**) Comparing nucleotide diversity at effector (red) and non-effector genes (turquoise) we observe again that wild-host effector diversity is higher than both wild-host non-effector gene diversity and agricultural effector diversity. Gene CDS regions polymorphic in one or other population are plotted (see gene n under boxplot). (**D**) Genetic differentiation ($F_{ST}$) plotted for gene regions (5 kb up- and down-stream) for both host-differentiated, and population-differentiated groups. The there is a strong positive correlation in genetic differentiation between both grouping methods (Host and DAPC). Host differentiated groups are lower on average. The inlayed plot displays all 272 non-effector genes that have a higher genetic differentiation for Host-differentiated groups than DAPC-differentiated, the relative difference is displayed in the colour gradient. There were zero effector genes that had higher differentiation in Host relative to DAPC groupings (see **Source data 1**).

