## [Editor Report]

This important study addresses the question of how pathogen reservoirs can play a role in the emergence of new virulences, using beets and Uromyces beticola as a pathosystem. The authors find convincing evidence for higher signals of diversity and differentiation in effector genes and significant differences in reproductive rates between two pathogen populations that occur either on wild or on wild and domesticated beets. These findings have the potential for developing approaches that reduce the evolution of pathogen resistance in crops.

---

## [Decision Letter]

[Editors' note: this paper was reviewed by Review Commons.]

Thank you for submitting your article "Developing a crop- wild-reservoir pathogen system to understand pathogen evolution and emergence" for consideration by *eLife*. Your article has been reviewed by 2 peer reviewers at Review Commons, and the evaluation at *eLife* has been overseen by me as Senior and Reviewing Editor.

I very much enjoyed reading your manuscript, which is very timely. I agree with you that it addresses an important blind spot in the field, and it is exciting to see the development of a new plant-pathosystem that includes both closely related, sympatric agricultural and wild hosts and closely related, sympatric agricultural and wild pathogens.

Based on your manuscript, the reviews, and your responses, we invite you to submit a revised version incorporating the revisions as outlined in your response to the reviews, as well as revisions addressing the following points:

- There are three weak points in the work:

(i) The reference genome is based on short reads. Given that generating long reads has become trivial (and cheap), I ask you to update the genome with a long-read assembly. If there are convincing reasons why you cannot generate a more contiguous assembly, please discuss this with me; you can reach me at weigel.*eLife*@gmail.com (note that I'm on vacation until the end of this week).

(ii) The number of isolates analyzed is relatively small, but given the striking differences you discovered between the agricultural and wild populations, this is acceptable. I would nevertheless ask you to explicitly discuss this potential weakness.

(iii) No simulations of demographic processes. Regardless of whether or not you can generate a long-read assembly, I would like you to look at recombination rates, given that you have contigs that extend over tens of kb. Hopefully, this will allow you to do the simulations the reviewers asked for. However, even if you cannot do simulations, I would like to learn whether there is any evidence for recombination rates being different between the two populations.

- Please remove Figure 3A from the main text. Sure, the difference between the two populations might be statistically significant, but given that the difference is so small and this simply reflects the large number of comparisons made, it does not really add information beyond what you show in Figure 3C.

[Editors' note: further revisions were suggested prior to acceptance, as described below.]

Thank you for submitting your article "Developing a crop- wild-reservoir pathogen system to understand pathogen emergence and evolution" for consideration by *eLife*. Your article has been reviewed by 1 peer reviewers at Review Commons, and the evaluation at *eLife* has been overseen by a Reviewing Editor and Detlef Weigel as the Senior Editor.

Based on the previous reviews and the revisions, the manuscript has been improved but there are some remaining issues that need to be addressed, as outlined below:

1. Polymorphism in Gene CDS Regions: Provide an explanation in the Discussion.

2. Impact of Reproductive Pathway on Effector Diversity: Include an overview of the reproductive pathways of rust fungi in the Discussion.

*Reviewer #2 (Recommendations for the authors):*

I would like to commend the authors on their comprehensive revision of the manuscript. The additional changes and the detailed explanations provided in response to the concerns raised in my previous review are much appreciated and have significantly improved the manuscript. I am satisfied with the progress made so far. However, I have two further suggestions that I believe would enhance the manuscript and help convey the findings more insightfully to readers.

1. Polymorphism in Gene CDS Regions:

In the Materials and methods section, the authors mention that approximately 50% of the gene coding sequences (CDS) did not show polymorphism. While this is an intriguing observation, it seems notably high. I believe this point is relevant to the editor's earlier comments regarding recombination rates in these populations. Given the authors' findings of higher levels of diversifying selection, particularly in effectors across both crop and wild accessions, it would be valuable to provide an explanation for this lack of polymorphism in the Discussion. This could help readers better understand the beet rust system and the evolutionary dynamics at play.

2. Impact of Reproductive Pathway on Effector Diversity:

The manuscript currently gives limited attention to the role of Uromyces beticola's reproductive pathway in shaping effector diversity across crop and wild hosts. This aspect is critical, especially when considering how reproductive modes influence pathogen diversity and the potential implications for managing crop-wild pathogen reservoirs under climate change. Including an overview of the reproductive pathways of rust fungi in the Discussion would provide valuable context for readers and strengthen the manuscript's relevance to the broader understanding of crop disease evolution.

---

## [Author Response]

General Statements

We are developing systems in which to address questions around crop pathogen adaptation and introgression. Crop systems represent a unique opportunity to understand pathogen adaptation because host diversity is restricted in a way that wild host diversity is not. This is particularly true where crops share pathogens with wild crop relatives. Work in this area is important for our understanding of host pathogen co-evolution as well as having applications in crop protection, pathogen surveillance and emergence prediction.

Here, we compare population genetic diversity in a (wild and crop) beet infecting rust pathogen. We find that a wild beet pathogen population has greater diversity than the crop infecting population and that effector genes are specifically impacted by this diversity. Moreover, effectors are more differentiated than expected (based on non-effector genes) and this is consistent with predictions based on the hypothesis that effector diversity is selected for by the host. We also find evidence that the crop pathogen favours a clonal mode of reproduction.

The reviewers have, rightly, highlighted that these analyses are preliminary. To the best of our knowledge this is the first genomic / population genomic research on this rust pathogen. The 600Mb rust assembly and annotation is a first, and large rust assemblies are known for difficulties in assembly. In addition, we have re-sequenced more than 40 isolates to begin to address questions considering crop pathogen adaptation. We thank the reviewers for their time and this opportunity to publish this work as we are keen to continue to build on the exciting signals we uncover.

Description of revisions

For consistency we combine reviewer’s comments into sections where we feel they address related points..

**Reviewer #1:**
Further work required on divergence to address the observed level of gene flow between wild and crop populations and the lack of admixed/hybrid genomes.Authors should plot Fst and Dxy along the main scaffolds. It would permit to see whether there is huge peaks of divergence and differentiation between the two populations.

We have measured Dxy across the genome and plotted against Fst across a number of contigs. This is part of a new supplement (8), where we emphasise how short these contigs are and that just a few genes on average are present in each one. We do see peaks and troughs in both *D_XY_* and *F_ST_* but we don’t feel we are able to plot data over a large enough distance to draw appropriate conclusions. We do plot *D_XY_* windows across effectors against non-effector genes and these confirm our previous observation based on *F_ST_* that divergence is greater at effector genes (see supplement 8).

Virulence can be mediated at the expression level by effector silencing and the reviewer suggests looking for premature stop codons, deletion of an exons, or mutations in the promoter region.Authors should also look for how many effector genes are non-expressed in cultivated population face to wild population.

SNPeff was used to identify and classify the impact of polymorphism across the genome. We queried this data in relation to effector and non-effector genes. We found that alternative splicing impacted non-effector genes to a greater degree. We also looked more generally at nonsense mutations and other High, then Medium, and Low impact variants (SNPs and indels combined). There were some instances where effectors in the wild population a greater number of these, but nothing that allowed us to draw a more general conclusion about the evolution of effectors, or evolution in reservoirs of diversity. However, this information may be useful for the community as analyses from crop pathogen reservoirs increase and are included in Supporting Information 8.

Simulations would provide stronger support for conclusions.

Absolutely. Simulations are important to validate our conclusions and improve our hypotheses as to the levels of selection, rates of gene flow and/or boom and bust. An important parameter for this model is the rate of recombination, and whether it varies between populations. In the main text of the MS we have utilised genotype (heterozygosity) based metrics and previously avoided linkage based metrics. This is because this is a repetitive, fragmented genome assembly, and in which the reference assembled contigs will represent a mosaic of two dikaryotic loci. Present analyses were designed to be informative at the population level, as the resource requirements are lower and therefore, the benefit to the community (generally lacking phased assemblies) are greater. At the request of the reviewer and editor we have added an analysis of recombination (and of selective sweeps, see below) to supplement 8.

Figure 2 network suggests strong divergence between populations, and this needs further exploration because divergence and the locus level is very low.– The reviewer requests a PCA– (Reviewer #2 missed the Machine Learning in the supplementary which also speaks to this work).– The Editor also suggested edits to improve clarity to figure 2

We have moved most Figure 2 panels to Figure 3. The suggested new PCA analysis now dominates Figure 2. The original Figure 2 (now 3) has reduced the content and includes only the map, the network, and the admixture plot. The new Figure 2 includes the expanded PCA and original DAPC analyses. By expanding PCA analyses, as suggested by reviewers, we can see in greater detail: -distinctions between isolates, -crop isolates tend to have long terminal branches for each isolate, -assignment to wild or crop population is high and not lower for the five wild harvested isolates identified within the crop infecting population.

We have also specifically mentioned in the main text the work that was done to try to partition those five individuals using machine learning methods. This work remains included in detail in Supporting Information 5.

Run Selscan ZA Szpiech and RD Hernandez (2014) or similar to look at indicators of selection.

We have run RAiSD which is a linkage based selective sweep analysis method and this work now features in the main text, ~ln 266. As outlined above and briefly in the main text, analyses relying on linkage are caveated by the lack of phasing in this repetitive dikaryotic assembly. In the main text we outline the short length of contigs, containing in most cases less than three genes. At the genome level we observed effectors above 95% *μ statistic* values and the effector genes we observed were not proportionately more than expected by chance (see main text). We did however observe that the *μ statistic* was higher on average across regions containing effectors than non-effector genes and we now present this in Figure 4.

Minor CommentsFigure 2 is somewhat hard to understand. There is too much data here. I would prefer a PCA plot, just showing strains plotted along the first axis and showing that there is no hybrid.

Figure 2 has been significantly altered. We have run a PCA (see also specific major comment) and plotted strains along both axes to show that the potential hybrids don’t even differentiate across the second axis.

Reviewer #2:Effector candidates were not evaluated/characterized in any form.Authors should compare pathogen features to other related species and try to contrast what stands out, especially the effectors' diversity

The reviewer referred to a statement in which we suggest that it is difficult to functionally annotate effectors (“According to the authors, it is difficult to functionally annotate these genes in general”). Our statement in the manuscript was not intended to suggest that we did not annotate them, only that, effectors less often receive a functional annotation (in the databases used to annotate them). This is likely a consequence of their rapid evolution. To clarify, we did use effector functional annotations to refer to the presence of shared effector annotations in other rust species. For example, In the Results section we highlighted effector functional annotations and conservation among the Pucciniales (e.g. Rust Transferred Protein, Α-amylase, CSEP-06 and PriA, among others). We have clarify this below and in the main text to reflect our efforts in that area.

Details of cross species functional annotation were included in Supporting Information 01. They included annotation using AHRD, UniProt (Swss-Prot and TrEMBL) blast and InterProScan. AHRD uses a database of unbiased ground truth set of high-quality protein annotations with minimal redundancy to assign GO annotations. UniProt is the world’s leading high-quality protein sequence and functional information. It contains more than 190 million sequences with which to assign functional annotations to proteins. InterProScan was used to assign proteins into families as well as predict domains. All these methods utilise cross species information to assign gene/domain function and ontology (GO), the output from these is included for each gene, in addition to that gene’s population genetic signature (Supporting Information 09).

Where exactly the machine learning was used?

Machine learning was used in the supplement to try to experiment with forcing a partition between isolates. This involved first using them, and then removing them from the training set. The models were not able to partition those wild infecting isolates that were partitioned within the crop population, unless they appeared in the training set.

Minor CommentsLines 184 – 186: Can the lack of admixture and gene low among these wild isolates also explain this observation? what about the levels of FIS in these isolates? Clonality in these populations may have a significant impact on the genetic diversity in these populations.

This point appropriately tries to dig further into the relationship between the two populations, as well as that between the five isolates that fall, counterintuitively, in the crop infecting population. This point was originally in reference to the Splitstree network, and we have added significant extra information to complement this which we feel backs up our original interpretation, including:

– A new PCA analysis and an NJ tree which isolate population affiliation has not changed in neither the PCA nor NJ tree. Isolates that split from the crop population in the PC2 contain just one of the five isolates in question. This one isolate falls within a group of crop infecting isolates which can also be seen in the NJ tree and are specifically mentioned in the figure legend. The NJ tree also has a different branching pattern in the two populations, again consistent with the observations the reviewer is suggesting we address (see F_IS_ below)

– sNMF analysis (now Figure 3) shows that the five isolates in question (asterisks) are no more partitioned (hybridised) than other crop isolates and this is now also complemented by a plot in figure 2 which looks at the probability of each individual being assigned to its population. These probabilities are all close to 1.

*–* We plot the levels of F_IS_ (now Figure 6) where we also highlight (with crosshairs) the five northern wild beet isolates that partition within the crop clade. These isolates are not distinct from this crop clade but the clade itself is distinct from the wild one where the signal of clonality is generally lower.

– We now highlight these five isolates with asterisks in figures, or in the text throughout so that the reader may see in each analysis that these isolates fall within the crop clade.

lines 216-220: Is this also reflected in the excess of heterozygosity non-effectors in these crop populations? The mutations should equally accumulate in both gene categories.

Thank you, and no, we now look at this just prior to the work on nucleotide diversity that the reviewer is referring to (line ~250). We find as expected that observed heterozygosity is greater in the wild population (than agri), but within population there is no significant difference in the observed level of heterozygosity at effectors and non-effector genes. The test for excess heterozygosity did show a skew in the distribution of p values towards increased significance (for both gene types) in the crop population. However, this did not significantly impact the numbers of genes for which the test crossed the 5% threshold (28 crop genes vs 22 wild genes, out of 8950 in total). We have not included this excess heterozygosity analysis in the text.

lines 219-220: it is not clear which CDS are being referred to here; Are you talking about the correlation between the CDSs of wild and crops or effectors and non-effectors?

Thank you, yes, this has now moved to line 274 but we have added that this is comparison between populations

Figure 1: I suggest separating F and G from the rest

Thank you for the suggestion. We have opted to keep these panels in figure 1 because they are part of the methodology from sampling, extraction and genome assembly.

Figure 3: D. Unless this is a noe to one window comparison of pi, this plot does not necessarily show a correlation. Please explain how the windows were treated in this comparison.

This is now Figure 4B. It is not a window-based comparison, it is as the reviewer suggests, a one to one calculation of nucleotide diversity value calculated from diversity within each gene CDS from each population.

Figure 4: A. I would expect a relatively high correlation between the FST and π in effectors. Does this include both wild and crop effectors?

Figure 4A, is diversity measured within genes across the whole experiment (both populations combined). We have added this to the legend text.

I spotted a number of typos throughout the manuscript. So I suggest the authors pay attention to punctuation and typos.

I apologise. I am particularly bad at spotting typos. I hope we have caught them this time around.

[Editors’ note: what follows is the authors’ response to the second round of review.]

I very much enjoyed reading your manuscript, which is very timely. I agree with you that it addresses an important blind spot in the field, and it is exciting to see the development of a new plant-pathosystem that includes both closely related, sympatric agricultural and wild hosts and closely related, sympatric agricultural and wild pathogens.Based on your manuscript, the reviews, and your responses, we invite you to submit a revised version incorporating the revisions as outlined in your response to the reviews, as well as revisions addressing the following points:- There are three weak points in the work:(ii) The number of isolates analyzed is relatively small, but given the striking differences you discovered between the agricultural and wild populations, this is acceptable. I would nevertheless ask you to explicitly discuss this potential weakness.

Addressed in the discussion in ‘Genomic tools for population genetics’, where we now also expand on other weaknesses including the fragmented assembly

(iii) No simulations of demographic processes. Regardless of whether or not you can generate a long-read assembly, I would like you to look at recombination rates, given that you have contigs that extend over tens of kb. Hopefully, this will allow you to do the simulations the reviewers asked for. However, even if you cannot do simulations, I would like to learn whether there is any evidence for recombination rates being different between the two populations.

Thank you. While many of our analyses are based around signals of effectors, recombination, is the main signal we want to pin down. We think its importance is overlooked in crop pathogen systems. We have run recombination analysis as suggested (LDhat) and it does find greater recombination in one population, although contrary to expectation, it is higher in the crop, not wild population. The results from genotype and linkage based methodologies contradict one another. We present both results and discuss the advantages of a more contiguous assembly. We also emphasize the importance of genotype based metrics for surveillance based methodologies. We will go on to improve this assembly (separately for wild and crop populations) and pin down recombination to develop our hypothesis using simulation, in further work.

- Please remove Figure 3A from the main text. Sure, the difference between the two populations might be statistically significant, but given that the difference is so small and this simply reflects the large number of comparisons made, it does not really add information beyond what you show in Figure 3C.

This has been removed and figures 2 and 3 have been significantly altered.

[Editors’ note: what follows is the authors’ response to the third round of review.]

Based on the previous reviews and the revisions, the manuscript has been improved but there are some remaining issues that need to be addressed, as outlined below:1. Polymorphism in Gene CDS Regions: Provide an explanation in the Discussion.2. Impact of Reproductive Pathway on Effector Diversity: Include an overview of the reproductive pathways of rust fungi in the Discussion.Reviewer #2 (Recommendations for the authors):I would like to commend the authors on their comprehensive revision of the manuscript. The additional changes and the detailed explanations provided in response to the concerns raised in my previous review are much appreciated and have significantly improved the manuscript. I am satisfied with the progress made so far. However, I have two further suggestions that I believe would enhance the manuscript and help convey the findings more insightfully to readers.1. Polymorphism in Gene CDS Regions:In the Materials and methods section, the authors mention that approximately 50% of the gene coding sequences (CDS) did not show polymorphism. While this is an intriguing observation, it seems notably high. I believe this point is relevant to the editor's earlier comments regarding recombination rates in these populations. Given the authors' findings of higher levels of diversifying selection, particularly in effectors across both crop and wild accessions, it would be valuable to provide an explanation for this lack of polymorphism in the Discussion. This could help readers better understand the beet rust system and the evolutionary dynamics at play.

We agree, at the CDS level diversity appears low and we now raise this observation in the Discussion (ln494), in relation to a point on longer term population fluctuations. Ultimately, we would like to look at the genome scale for an association between polymorphism and recombination. We feel this would begin to address the reviewer’s point. However, the current fragmented nature of the genome makes this impossible. This could be addressed using long read assemblies, funded by a new grant hopefully aided by this publication. The window-based recombination analyses we have added in response to the previous review are already reaching the limits of what can be supported (in our opinion), and we don’t think it appropriate to look for associations between diversity – effectors – recombination, at this stage. We have already highlighted in the discussion, our requirement for a longer phased reference assembly in order to begin to address the kinds of questions.

2. Impact of Reproductive Pathway on Effector Diversity:The manuscript currently gives limited attention to the role of Uromyces beticola's reproductive pathway in shaping effector diversity across crop and wild hosts. This aspect is critical, especially when considering how reproductive modes influence pathogen diversity and the potential implications for managing crop-wild pathogen reservoirs under climate change. Including an overview of the reproductive pathways of rust fungi in the Discussion would provide valuable context for readers and strengthen the manuscript's relevance to the broader understanding of crop disease evolution.

Thank you. We are sorry to hear that the value of reproductive pathway hasn’t come across. We agree with the reviewer and now identify specific life stages in the introduction (ln110). We’ve also made changes to the discussion, in several places under the subheading “Partitioning modes of reproduction”.